# RATE-ADAPTIVE QUANTIZATION: A MULTI-RATE CODEBOOK ADAPTATION FOR VECTOR QUANTIZED MODELS

## ABSTRACT

Learning discrete representations with vector quantization (VQ) has emerged as a powerful approach in representation learning across vision, audio, and language. However, most VQ models rely on a single, fixed-rate codebook, requiring extensive retraining for new bitrates or efficiency requirements. We introduce *Rate-Adaptive Quantization (RAQ)*, a multi-rate codebook adaptation framework for VQ models. RAQ integrates a lightweight sequence-to-sequence (Seq2Seq) codebook generator with the base VQ model, enabling on-demand codebook adaptation to any target size at inference. Additionally, we provide a clustering-based post-hoc alternative for pre-trained VQ models, suitable when modifying the training pipeline or joint training is not feasible. Our experiments demonstrate that RAQ performs effectively across multiple rates and VQ models, often outperforming fixed-rate baselines. This model-agnostic adaptability enables a single system to meet varying bitrate requirements in reconstruction and generation tasks.

## 1 INTRODUCTION

Vector quantization (VQ) (Gray, 1984) is a fundamental technique for learning discrete representations for various tasks (Krishnamurthy et al., 1990; Gong et al., 2014; Van Niekerk et al., 2020) in the field of machine learning. The Vector Quantized Variational Autoencoder (VQ-VAE) (Van Den Oord et al., 2017; Razavi et al., 2019), which extends the encoder-decoder structure of the Variational Autoencoder (VAE) (Kingma & Welling, 2013; Rezende & Viola, 2018), introduces discrete latent representations that have proven effective across vision (Razavi et al., 2019; Esser et al., 2021), audio (Dhariwal et al., 2020; Yang et al., 2023), and speech tasks (Kumar et al., 2019; Xing et al., 2023). The inherently discrete nature of these modalities makes VQ particularly well-suited for complex inference and generation.

Recent developments have further enhanced VQ-based discrete representation learning by integrating it with deep generative models, such as Generative Adversarial Networks (GANs) (Esser et al., 2021) and Denoising Diffusion Probabilistic Models (DDPMs) (Cohen et al., 2022; Gu et al., 2022; Yang et al., 2023). As VQ models are integrated into these diverse generative frameworks, their utility and applicability in various tasks are becoming increasingly evident. However, even with these advancements, *scalability* remains a bottleneck. In practical settings such as live streaming, telepresence, and on-device applications, the available bandwidth and compute resources can fluctuate dramatically. A single fixed-rate VQ model either wastes bits when higher quality is possible or severely degrades fidelity under tight constraints. Maintaining separate VQ models for each bitrate is infeasible and incurs significant overhead. Hence, a robust framework that can seamlessly adapt its compression rate is crucial for real-world deployments.

Several works have explored enhancing the flexibility of codebooks. Li et al. (2023) introduced a codebook-resizing technique for publicly available VQ models by applying hyperbolic embeddings, Malka et al. (2023) propose a nested codebook to support multiple quantization levels, and multi-codebook vector quantization is used in speech (Guo et al., 2022) and a knowledge distillation setting (Guo et al., 2023). Recently, Huijben et al. (2024) focused on unsupervised codebook generation based on residual quantization by studying the vector quantizer itself. However, it remains impractical to increase the rate of an already-deployed quantizer by appending new residual

stages after training. Simply adding more codebooks tends to disrupt the learned latent distribution and often necessitates a reduction in the spatial/temporal resolution of feature maps. This post-hoc capacity-scaling bottleneck is what motivated the development of our RAQ framework, which adapts bit rates through a lightweight Seq2Seq module that generates new codebook embeddings, while leaving the original VQ architecture untouched.

In this paper, we present *Rate-Adaptive Quantization (RAQ)*, a framework designed to flexibly modulate the effective codebook size of a single VQ model without retraining. By incorporating a Sequence-to-Sequence (Seq2Seq) (Sutskever et al., 2014) into the VQ model, RAQ enables one system to cover multiple compression levels, reducing the need for separate models dedicated to each rate. This adaptability not only minimizes storage and maintenance costs but also provides a smoother user experience in real-time communications or streaming environments, where bandwidth availability can vary from moment to moment. While our main focus is on the Seq2Seq-based RAQ, we additionally propose a model-based alternative that applies differentiable $k$-means (DKM) (Cho et al., 2021) clustering to a pre-trained VQ model, offering codebook adaptation when joint training or architectural modification is not feasible. This simple approach provides a practical fallback in scenarios where retraining or model modification is not feasible.

Our contributions are summarized as follows:

- We propose the *Rate-Adaptive Quantization (RAQ)* framework for flexible multi-rate codebook adaptation, using a Sequence-to-Sequence (Seq2Seq) module to generate codebook embeddings of any target size without retraining. This method can be integrated into existing VQ models with minimal modifications.

- To mitigate distribution mismatch in autoregressive Seq2Seq codebook adaptation, we introduce a *cross-forcing* training procedure. This approach ensures stable codebook generation across diverse rates and enhances reconstruction fidelity.

- We evaluate RAQ on several VQ benchmarks and show that a *single* RAQ-enabled model consistently meets or exceeds the performance of multiple fixed-rate VQ baselines while using the same encoder-decoder architecture.

## 2 BACKGROUND

### 2.1 VECTOR-QUANTIZED VARIATIONAL AUTOENCODER

VQ-VAEs (Van Den Oord et al., 2017) can successfully represent meaningful features that span multiple dimensions of data space by discretizing continuous latent variables to the nearest codebook vector in the codebook. In a VQ model, learning of discrete representations is achieved by quantizing the encoded latent variables to their nearest neighbors in a trainable codebook and decoding the input data from the discrete latent variables. To represent the data $\mathbf{x}$ from dataset $\mathcal{D}$ discretely, a codebook $\mathbf{e}$ consisting of $K$ learnable codebook vectors $\{e_i\}_{i=1}^{K} \subset \mathbb{R}^d$ is employed (where $d$ denotes the dimensionality of each codebook vector $e_i$). The quantized discrete latent variable $\mathbf{z}_q(\mathbf{x}|\mathbf{e})$ is decoded to reconstruct the data $\mathbf{x}$. The vector quantizer $Q$ is modeled as a deterministic categorical posterior that maps each spatial position $[m, n]$ of the continuous latent representation $\mathbf{z}_e(\mathbf{x})[m, n]$ of the data $\mathbf{x}$ by a deterministic encoder $f_\phi$ to $\mathbf{z}_q(\mathbf{x}|\mathbf{e})[m, n]$ by finding the nearest neighbor from the codebook $\mathbf{e} = \{e_i\}_{i=1}^{K}$ as

$$\mathbf{z}_q(\mathbf{x}|\mathbf{e})[m, n] = Q\Big(\mathbf{z}_e(\mathbf{x})[m, n]\big|\mathbf{e}\Big) = \arg\min_i \|\mathbf{z}_e(\mathbf{x})[m, n] - e_i\|, \tag{1}$$

The quantized representation uses $\log_2 K$ bits to index one of the $K$ selected codebook vectors $\{e_i\}_{i=1}^{K}$. The deterministic decoder $f_\theta$ reconstructs the data $\mathbf{x}$ from the quantized discrete latent variable $\mathbf{z}_q(\mathbf{x}|\mathbf{e})$ as $\hat{\mathbf{x}} = f_\theta\big(\mathbf{z}_q(\mathbf{x}|\mathbf{e})|\mathbf{e}\big)$. During the training process, the encoder $f_\phi$, decoder $f_\theta$, and codebook $\mathbf{e}$ are jointly optimized to minimize the loss $\mathcal{L}_{\text{VQ}}\big(\phi, \theta, \mathbf{e}; \mathbf{x}\big) =$

$$\underbrace{\log p_\theta(\mathbf{x}|\mathbf{z}_q(\mathbf{x}|\mathbf{e}))}_{\mathcal{L}_{\text{recon}}} + \underbrace{\big|\big|\text{sg}\big[f_\phi(\mathbf{x})\big] - \mathbf{z}_q(\mathbf{x}|\mathbf{e})\big|\big|_2^2}_{\mathcal{L}_{\text{embed}}} + \underbrace{\beta\big|\big|\text{sg}\big[\mathbf{z}_q(\mathbf{x}|\mathbf{e})\big] - f_\phi(\mathbf{x})\big|\big|_2^2}_{\mathcal{L}_{\text{commit}}} \tag{2}$$

where $\text{sg}[\cdot]$ is the *stop-gradient* operator. The $\mathcal{L}_{\text{recon}}$ is the reconstruction loss between the input data $\mathbf{x}$ and the reconstructed decoder output $\hat{\mathbf{x}}$. The two $\mathcal{L}_{\text{embed}}$ and $\mathcal{L}_{\text{commit}}$ losses apply only to codebook variables and encoder weights with a weighting hyperparameter $\beta$ to prevent fluctuations from

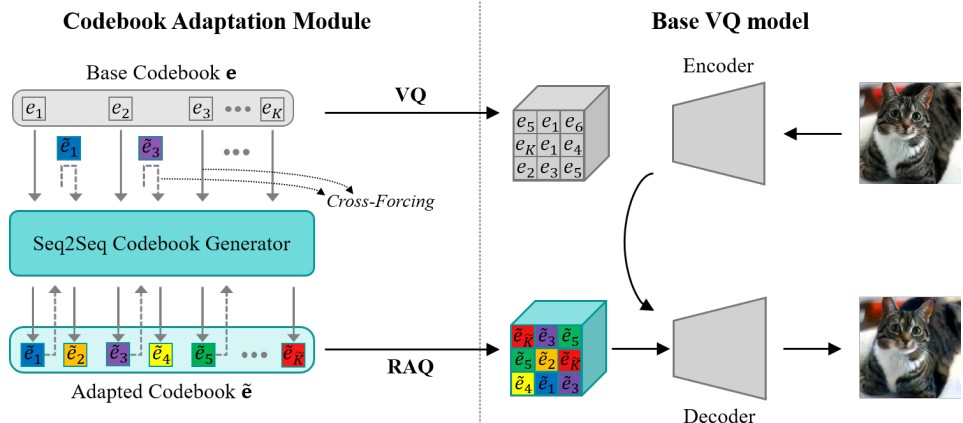

Figure 1: An overview of our RAQ framework applied to a base VQ model. During training, the codebook adaptation module employs cross-forcing to generate adapted codebooks $\tilde{\mathbf{e}}$ for randomly sampled sizes $\widetilde{K}$ from the base codebook $\mathbf{e}$. At inference, a user-specified target $\widetilde{K}$ produces the corresponding adapted codebook for rate-adaptive quantization.

one codebook vector to another. Since the quantization process is non-differentiable, the codebook loss is typically approximated via a straight-through gradient estimator (Bengio et al., 2013), such as $\partial \mathcal{L} / \partial f_\phi(\mathbf{x}) \approx \partial \mathcal{L} / \partial \mathbf{z}_q(\mathbf{x})$. Both conventional VAE (Kingma & Welling, 2013) and VQ-VAE (Van Den Oord et al., 2017) have objective functions consisting of the sum of reconstruction error and latent regularization. To improve performance and convergence rate, an exponential moving average (EMA) update is usually applied for the codebook optimization (Van Den Oord et al., 2017; Razavi et al., 2019). Thus, VQ models serve as a foundation for many advanced generative models, forming the core approach to discrete latent representation.

## 2.2 SEQUENCE-TO-SEQUENCE LEARNING

The Seq2Seq (Sutskever et al., 2014) model is widely used in sequence prediction tasks such as language modeling and machine translation (Dai & Le, 2015; Luong et al., 2016; Ranzato et al., 2016). The model employs an initial LSTM, called the encoder, to process the input sequence $x_{1:N}$ sequentially and produce a substantial fixed-dimensional vector representation, called the context vector. The output sequence $y_{1:T}$ is then derived by a further LSTM, the decoder. A Seq2Seq with parameters $\psi$ estimates the distribution of output sequence $y_{1:T}$ by decomposing it into an ordered product of conditional probabilities:

$$p\big(y_{1:T}|x_{1:N};\psi\big) = \prod_{t=1}^{T} p\big(y_t|y_{1:t-1}, x_{1:N};\psi\big) \tag{3}$$

During training, the Seq2Seq model typically uses *teacher-forcing* (Williams & Zipser, 1989), where the target sequence is provided to the decoder at each time step, instead of the decoder using its own previous output as input. This method helps the model converge faster by providing the correct context during training.

## 3 METHODS

Although VQ models have demonstrated strong performance across modalities, their fixed codebook size can limit adaptability under varying data characteristics or resource constraints. In practice, the choice of the codebook size $K$ (a key hyperparameter in VQ models) can vary widely with (i) on the application domain (e.g., small codebooks, $K \approx 8$, in certain audio domains (Chae et al., 2025)), (ii) input dimensionality and resolution (e.g., image generation models report $K$ from $512$ up to $16{,}384$ (Esser et al., 2021)), and (iii) the model architecture and quantization scheme. This variability often forces practitioners to retrain or maintain multiple VQ models at different rates. To address these challenges, we introduce the RAQ framework, which adjusts a VQ model's rate

by increasing or decreasing the codebook size $K$ on demand. We formalize RAQ as a mapping $\Psi : \mathbb{R}^{d \times K} \longrightarrow \mathbb{R}^{d \times \widetilde{K}}$ for any integer $\widetilde{K} \in \mathbb{N}$. We next detail a Seq2Seq-based RAQ strategy, followed by a model-based (clustering) alternative.

## 3.1 RATE-ADAPTIVE QUANTIZATION

**Overview** The RAQ framework is designed to integrate seamlessly with existing VQ models without requiring significant architectural modifications. As illustrated in Figure 1, RAQ is integrated into a base VQ model which consists of an encoder-decoder pair and a trainable base codebook $\mathbf{e}$. The adapted codebook $\tilde{\mathbf{e}}$ is generated by a Seq2Seq model from the base codebook $\mathbf{e}$. During training, $\widetilde{K}$ is randomly sampled under a cross-forcing strategy. At inference, a user-specified target $\widetilde{K}$ is used to generate the codebook on demand.

**Scope and Compatibility with Vector Quantizers** RAQ exclusively operates at the quantization layer of existing VQ models and applies to any quantizer that uses a vector-embedding-based discrete representation. Hierarchical VQ (Razavi et al., 2019), stochastic quantization (Takida et al., 2022), residual quantization (Huijben et al., 2024), and linear-transformed VQ (Zhu et al., 2024) are some of the representative examples. Importantly, RAQ does not alter the base training procedure or the overall architecture. Unlike autoregressive token predictors (Esser et al., 2021; Yu et al., 2022; Huijben et al., 2024), which generate discrete index sequences under fixed codebook embeddings, RAQ synthesizes the codebook itself.

**Autoregressive Generation of Adapted Codebooks** Our codebook adaptation module $G_\psi$ maps a base codebook $\mathbf{e}$ to an adapted codebook $\tilde{\mathbf{e}}$ via an Seq2Seq model. Each base codebook vector $e_i$ of $\mathbf{e}$ is treated analogously to a token in language modeling. We train the Seq2Seq module to produce a set of $\widetilde{K}$ adapted codebook embeddings. This autoregressive generation ensures that each $\tilde{e}_i$ is conditioned on the previously generated vectors, promoting coherence and structural consistency. Formally, the adapted codebook is generated as

$$p(\tilde{\mathbf{e}}|\mathbf{e}; \psi) = \prod_{i=1}^{\widetilde{K}} p(\tilde{e}_i | \tilde{e}_{<i}, e_{1:K}; \psi) \tag{4}$$

where $\tilde{e}_{<i}$ denotes the vectors generated before step $i$. Unlike typical Seq2Seq setups that optimize next-token likelihood, the order of $\tilde{e}_{i:\widetilde{K}}$ is not semantically meaningful here; what matters is the distribution of embeddings. Accordingly, RAQ trains the Seq2Seq module with the base VQ objective computed using $\tilde{\mathbf{e}}$ (e.g., reconstruction or perceptual losses), tying codebook generation directly to downstream reconstruction quality without imposing an arbitrary sequence order or introducing extra losses.

**Codebook Encoding** We begin by initializing the target codebook size $\widetilde{K}$. During training, $\widetilde{K}$ is randomly sampled from a predefined range at each iteration. Each base codebook vector $e_i$ is sequentially processed by LSTM cells, whose hidden and cell states $(\boldsymbol{h}, \boldsymbol{c})$ summarize context over the base codebook. This encoding captures dependencies among the base embeddings and provides a foundation for generating a coherent adapted codebook.

**Codebook Decoding via Cross-Forcing** We decode with a *cross-forcing* strategy that alternates teacher forcing and free running strategy to stably generate variable-size codebooks. Standard teacher forcing (Williams & Zipser, 1989) can be brittle when the target adapted codebook $\tilde{\mathbf{e}}$ is much longer than the base codebook, amplifying exposure bias. Cross-forcing mitigates this by interleaving the two modes, akin to professor forcing (Lamb et al., 2016). During training, the cross-forcing strategy operates as:

- **Teacher-Forcing Phase:** For odd indices $i$ up to $2K$ (i.e., $1 \leq i \leq 2K$ and $i$ is odd), the model uses the corresponding base codebook vector $e_j$ as input, where $j = \frac{i+1}{2}$:

$$\tilde{e}_i = \text{LSTM}_\psi(\tilde{e}_{<i}, e_j, \boldsymbol{h}, \boldsymbol{c}).$$

This ensures that the fundamental distributional features of the base codebook are preserved during the early generation steps.

- **Free-Running Phase:** For even indices $i$ up to $2K$ ($i$ is even and $i \leq 2K$), and for all indices beyond $2K$ (i.e., $i > 2K$), the model relies on its previously generated adapted codebook vector $\tilde{e}_{i-1}$:
$$\tilde{e}_i = \text{LSTM}_\psi(\tilde{e}_{<i}, \tilde{e}_{i-1}, \boldsymbol{h}, \boldsymbol{c}).$$
By switching to its own outputs, the model learns to maintain coherence and consistency across the adapted codebook vectors for different sizes of $\widetilde{K}$.

Learning the codebook adaptation module $G_\psi$ via cross-forcing is a key component of our RAQ. We provide an empirical evaluation of its effectiveness in Appendix A.3.2.

**Training Procedure**  RAQ follows the objective of the base VQ model. Concretely, for a conventional VQ-VAE, let $\mathcal{L}_{\text{VQ}}$ (in equation 2) denote the standard loss. At each iteration, we sample a target size $\widetilde{K}$ from a predefined range, generate an adapted codebook $\tilde{\mathbf{e}} = G_\psi(\mathbf{e}; \widetilde{K})$, and jointly optimize $(\phi, \theta, \mathbf{e}, \psi)$ by minimizing a combined objective $(\mathcal{L}_{\text{VQ}} + \mathcal{L}_{\text{RAQ}})$, where $\mathcal{L}_{\text{RAQ}}(\phi, \theta, \psi, \mathbf{e}; \mathbf{x}) =$

$$\log p_\theta\big(\mathbf{x}|\mathbf{z}_q(\mathbf{x}|G_\psi(\mathbf{e}))\big) + \big|\big|\text{sg}\left[f_\phi(\mathbf{x})\right] - \mathbf{z}_q\big(\mathbf{x}|G_\psi(\mathbf{e})\big)\big|\big|_2^2 + \beta\big|\big|\text{sg}\left[\mathbf{z}_q\big(\mathbf{x}|G_\psi(\mathbf{e})\big)\right] - f_\phi(\mathbf{x})\big|\big|_2^2. \quad (5)$$

Equivalently, RAQ trains by plugging the adapted codebook $\tilde{\mathbf{e}}$ into the existing objective function (no additional auxiliary losses are introduced). Sampling $\widetilde{K}$ each iteration exposes the model to multiple rates within a single training run.

## 3.2 Model-Based Rate-Adaptive Quantization (Alternative)

We present a post-hoc, model-based RAQ variant that adapts the codebook rate while leaving the rest of the base VQ model unchanged. Unlike the Seq2Seq-based RAQ, this approach adds no learnable modules; it directly resizes a pre-trained codebook $\mathbf{e}$ to a target $\widetilde{K}$ via clustering. Optional brief fine-tuning with the adapted codebook can be applied but is not required.

To obtain an adapted codebook $\tilde{\mathbf{e}}$ of size $\widetilde{K}$, we employ differentiable $k$-means (DKM) (Cho et al., 2021), originally proposed for compressing model weights via layer-wise clustering. Here, DKM is repurposed to cluster the embedding vectors in $\mathbf{e}$, yielding a reduced (or increased) codebook while preserving structure in the embedding space. We also leverage inverse functionalization (IKM) to accommodate increases in codebook size, enabling both rate reduction and rate expansion.

**Codebook Reduction** $(\widetilde{K} < K)$  In the rate-reduction task, DKM performs iterative, differentiable codebook clustering on $\widetilde{K}$ clusters. Let $\mathbf{C} = \{c_j\}_{j=1}^{\widetilde{K}}$ be the cluster centers for the base codebook $\mathbf{e}$ (Further details are provided in Appendix A.2.1). The process is as follows:

- Initialize the centroids $\mathbf{C} = \{c_j\}_{j=1}^{\widetilde{K}}$ by randomly selecting $\widetilde{K}$ codebook vectors from $\mathbf{e}$ or by using $k$-means++. The last updated $\mathbf{C}$ is used in subsequent iterations.
- Compute the Euclidean distance between each $e_i$ and $c_j$, denoting $D_{i,j} = -f(e_i, c_j)$ to form the matrix $\boldsymbol{D}$.
- Form the attention matrix $\boldsymbol{A}$ via a softmax with temperature $\tau$, where each row satisfies
$$A_{i,j} = \frac{\exp\left(\frac{D_{i,j}}{\tau}\right)}{\sum_k \exp\left(\frac{D_{i,k}}{\tau}\right)}.$$
- Compute the candidate centroids $\widetilde{\mathbf{C}} = \{\tilde{c}_j\}$ by $\tilde{c}_j = \frac{\sum_i A_{i,j} e_i}{\sum_i A_{i,j}}$, then update $\mathbf{C} \leftarrow \widetilde{\mathbf{C}}$.
- Repeat until $\|\mathbf{C} - \widetilde{\mathbf{C}}\| \leq \epsilon$ or the iteration limit is reached. We then multiply $\boldsymbol{A}$ by $\mathbf{C}$ to obtain the final $\tilde{\mathbf{e}}$.

The above iterative process can be summarized as follows:

$$\tilde{\mathbf{e}} = \underset{\tilde{\mathbf{e}}}{\arg\min}\, \mathcal{L}_{\text{DKM}}(\mathbf{e}; \tilde{\mathbf{e}}) = \underset{\mathbf{C}}{\arg\min}\, |\mathbf{C} - \boldsymbol{A}\mathbf{C}| = \underset{\mathbf{C}}{\arg\min} \sum_{j=1}^{\widetilde{K}} \left| c_j - \frac{\sum_i A_{i,j} e_i}{\sum_i A_{i,j}} \right|. \quad (6)$$

Since the procedure is differentiable, the centroids and soft assignments can be optimized with a few steps of SGD. The temperature $\tau$ controls assignment hardness. After convergence, we assign each codebook vector to its nearest centroid according to the last attention matrix $\boldsymbol{A}$, thereby finalizing the compressed codebook.

**Codebook Expansion** $(\widetilde{K} > K)$   As bandwidth and quality budgets increase, models benefit from a larger codebook, so codebook expansion must be supported. We propose an inverse functional DKM (IKM) method that grows the number of centroids from $K$ to $\widetilde{K}$ and then refines them using the same DKM updates used in reduction. Algorithmic details and optimization choices are deferred to Appendix A.2.2.

Model-based RAQ operates directly on the codebook of any pre-trained VQ model, enabling post-hoc codebook rate adjustment without introducing new learnable modules. This approach is useful when training or integrating Seq2Seq-based RAQ is impractical. As it relies on differentiable clustering, fine-tuning is also supported via post-training. Appendix A.2.4 provides comparisons with alternative clustering methods and the effect of post-training.

## 4    RELATED WORK

**VQ and its Improvements**   The VQ-VAE (Van Den Oord et al., 2017) has inspired numerous developments since its inception. Łańcucki et al. (2020) and Zheng & Vedaldi (2023) proposed codebook reset and online clustering methods to mitigate *codebook collapse*, improving training efficiency. SQ-VAE (Takida et al., 2022) incorporated stochastic quantization and a trainable posterior categorical distribution to enhance VQ performance. Recently, SimVQ (Zhu et al., 2024) tackles the long-standing representation-collapse issue in vanilla VQ models by reparameterizing the codebook via linear transformation. Several works have introduced substantial structural changes to VQ modeling; for instance, RQ-VAE (Lee et al., 2022) employed a two-step residual quantization framework for high-resolution images, while FSQ (Mentzer et al., 2023) replaced VQ with finite scalar quantization to address codebook collapse. Our approach focuses on making rate-adaptive VQ without substantially altering the quantization mechanism or architecture, allowing it to scale effectively in both basic and advanced VQ models.

**Variable-Rate Neural Image Compression**   Several studies have proposed variable-rate image compression based on autoencoders and VAEs (Yang et al., 2020; Choi et al., 2019; Cui et al., 2020), or using recurrent neural networks (Johnston et al., 2018). Song et al. (2021) introduced spatial feature transforms for compression, while Duong et al. (2023) combined learned transforms and entropy coding in a single model aligned with the rate-distortion curve. Variable-rate methods for discrete representation also exist. Dieleman et al. (2021) learn event-based codes with scalar quantization and control rate via channel budgets and target event rates, rather than resizing a fixed VQ codebook. In contrast, RAQ targets the VQ codebook itself. We adapt the vector embedding set to a user-specified size at inference without redesigning the underlying quantization mechanism or training a separate model per rate. Our RAQ directly addresses the practical need for a single VQ model to cover multiple bitrates.

## 5    EXPERIMENTS

### 5.1    EXPERIMENTAL SETUP

**Settings**   We perform empirical evaluations on 3 vision datasets: CIFAR10 (Krizhevsky et al.) and CelebA (Liu et al., 2015), and ImageNet (Russakovsky et al., 2015). We use the same architecture and hyperparameters within each baseline model. The adapted codebook sizes range from 16 to 1024 for CIFAR10, 32 to 2048 for CelebA, and 32 to 4096 for ImageNet, while the base codebooks of baseline VQ models are fixed. RAQ-based models set the base codebook size to the middle of the range. We also provide details on each model's parameter count and complexity in Appendix A.2.5.

**Evaluation Metrics**   We quantitatively evaluated our method using Peak-Signal-to-Noise-Ratio (PSNR), Structural Similarity Index Measure (SSIM), reconstructed Fréchet Inception Distance (rFID) (Heusel et al., 2017), Learned Perceptual Image Patch Similarity (LPIPS) (Zhang et al., 2018), and codebook perplexity. PSNR measures the ratio between the maximum possible power of a signal and the power of the corrupted noise affecting data fidelity (Korhonen & You, 2012). SSIM assesses structural similarity between two images (Wang et al., 2004). rFID and LPIPS assess the quality of reconstructed images by comparing the distribution of features extracted from

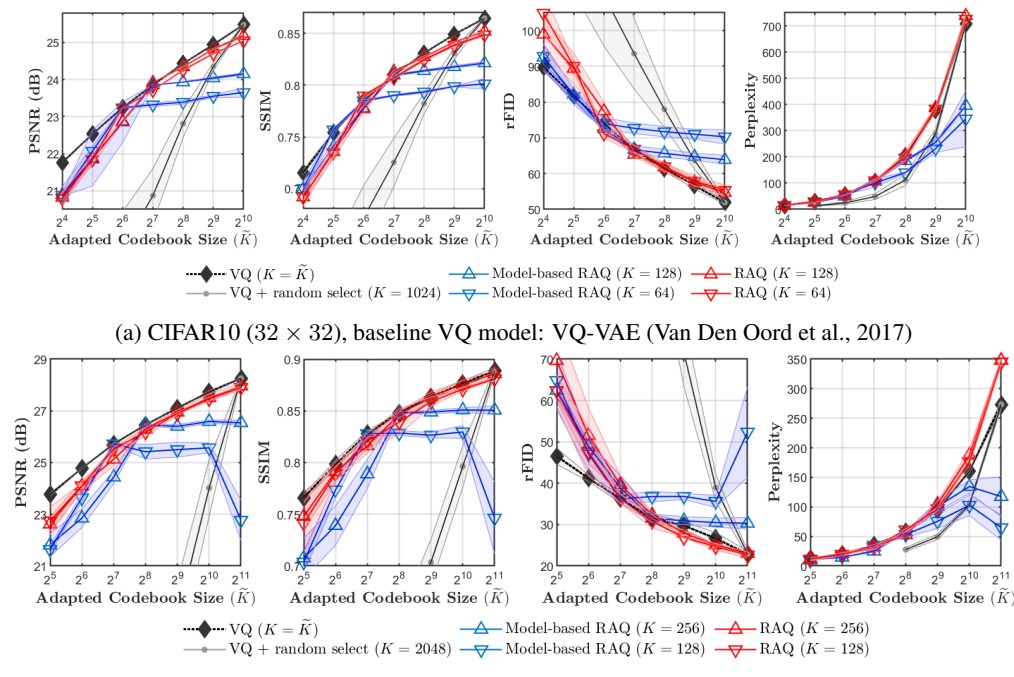

(a) CIFAR10 ($32 \times 32$), baseline VQ model: VQ-VAE (Van Den Oord et al., 2017)

(b) CelebA ($64 \times 64$), baseline model: VQ-VAE (Van Den Oord et al., 2017)

Figure 2: **Reconstruction results** on (a) CIFAR10 and (b) CelebA at various codebook sizes $\widetilde{K}$. The shaded area indicates the 95.45% confidence interval based on 4 runs with different seeds.

the test data with that of the original data. Codebook perplexity, defined as $e^{-\sum_i^{\widetilde{K}} p_{e_i} \log p_{e_i}}$ where $p_{e_i} = \frac{N_{e_i}}{\sum_j^{\widetilde{K}} N_{e_j}}$ and $N_{e_i}$ represents the encoded number for latent representation with codebook $e_i$, indicates a uniform prior distribution when the perplexity value reaches the codebook size $\widetilde{K}$.

## 5.2 QUANTITATIVE EVALUATION

We first empirically evaluate the effectiveness of RAQ using the vanilla VQ-VAE (Van Den Oord et al., 2017) on CIFAR10 and CelebA dataset. To establish a robust baseline, we trained multiple VQ-VAE models with varying codebook sizes $K$ as fixed-rate benchmarks. We then tested RAQ's adaptability by dynamically adjusting the codebook size $\widetilde{K}$ within a single VQ-VAE. Figure 2 shows the comparative results.

RAQ closely matches the performance of multiple fixed-rate VQ-VAE models across most metrics, demonstrating its ability to maintain high reconstruction quality while offering rate flexibility. Specifically, under identical compression rates and network architectures, all RAQ variants achieve PSNR and SSIM scores nearly on par with their fixed-rate counterparts (e.g., within about 0.94 dB difference in PSNR on CIFAR10). When we increase the rate, RAQ occasionally shows slightly lower PSNR and SSIM but improves rFID, reflecting improved perceptual quality and better alignment with the dataset distribution. For instance, at $\widetilde{K} = 512$ on CelebA, rFID improves by up to about 9.6% compared to a fixed-rate VQ-VAE with the same codebook size, highlighting RAQ's ability to maintain visual coherence and realism in generative tasks. However, the model-based RAQ variant typically underperforms our Seq2Seq-based approach except in certain rate-reduction tasks. For further discussion of model-based RAQ, see Section 5.4.

We also evaluate RAQ across four representative VQ baselines (hierarchical VQ-VAE-2 (Razavi et al., 2019), stage-1 VQGAN (Esser et al., 2021), SQ-VAE (Takida et al., 2022), and SimVQ (Zhu et al., 2024)) to assess both rate flexibility and architectural generality. Table 1 reports mean PSNR, SSIM, LPIPS, and codebook perplexity over four runs. RAQ maintains or improves reconstruction quality relative to fixed-rate baselines while enabling on-demand adaptation of the codebook size.

On VQ-VAE-2, RAQ increases perplexity at higher rates while yielding comparable or better PSNR/LPIPS at mid–high $\widetilde{K}$. On stage-1 VQGAN, RAQ improves LPIPS in the mid-rate regime

Table 1: Performance comparison for *VQ-VAE-2* (Razavi et al., 2019), *stage-1 VQGAN* (Esser et al., 2021), *SQ-VAE* (Takida et al., 2022), and *SimVQ* (Zhu et al., 2024) with and without RAQ at multiple adapted codebook sizes $\widetilde{K}$. **Bold** indicates that the proposed method outperforms the baseline model, and the † denotes that the improvement is statistically indistinguishable from the baseline based on overlapping 95.45% confidence intervals.

| Method | $K$ | $\widetilde{K}$ | PSNR↑ | SSIM↑ | LPIPS↓ | Prplx.↑ | Method | $K$ | $\widetilde{K}$ | PSNR↑ | SSIM↑ | LPIPS↓ | Prplx.↑ |
|---|---|---|---|---|---|---|---|---|---|---|---|---|---|---|
| | | | **VQ-VAE-2** / CelebA (128 × 128) | | | | | | | **VQGAN** / ImageNet (256 × 256) | | | |
| VQ-VAE-2 | 2048 | – | 33.37 | 0.9884 | 0.1050 | 334.7 | VQGAN | 512 | – | 21.00 | 0.7543 | 0.1000 | 296.3 |
| VQ-VAE-2 | 1024 | – | 32.73 | 0.9865 | 0.1172 | 183.2 | VQGAN | 256 | – | 20.65 | 0.7383 | 0.1083 | 140.0 |
| VQ-VAE-2 | 512 | – | 32.18 | 0.9842 | 0.1313 | 103.1 | VQGAN | 128 | – | 20.29 | 0.7221 | 0.1176 | 75.0 |
| VQ-VAE-2 | 256 | – | 31.29 | 0.9810 | 0.1464 | 61.6 | VQGAN | 64 | – | 19.92 | 0.7015 | 0.1303 | 40.1 |
| VQ-VAE-2 | 128 | – | 30.72 | 0.9780 | 0.1588 | 36.3 | VQGAN | 32 | – | 19.58 | 0.6834 | 0.1415 | 21.9 |
| VQ-VAE-2 + random select | 2048 | 1024 | 29.23 | 0.9717 | 0.1694 | 178.1 | VQGAN + random select | 512 | 256 | 19.98 | 0.7311 | 0.1185 | 148.5 |
| | | 512 | 28.01 | 0.9642 | 0.2067 | 103.0 | | | 128 | 18.93 | 0.7035 | 0.1444 | 74.1 |
| | | 256 | 26.43 | 0.9514 | 0.2584 | 60.4 | | | 64 | 17.37 | 0.6614 | 0.1917 | 37.2 |
| | | 128 | 16.21 | 0.7266 | 0.5536 | 14.1 | | | 32 | 14.64 | 0.5801 | 0.3148 | 18.9 |
| VQ-VAE-2 + **model-based RAQ** | 2048 | 1024 | 31.62 | 0.9813 | 0.1404 | 131.1 | VQGAN + **model-based RAQ** | 512 | 256 | 20.43 | 0.7381 | 0.1119 | 147.1 |
| | | 512 | 30.23 | 0.9733 | 0.1739 | 53.8 | | | 128 | 19.74 | 0.7182 | 0.1314 | 75.3 |
| | | 256 | 29.09 | 0.9658 | 0.2068 | 30.6 | | | 64 | 18.82 | 0.6908 | 0.1642 | 39.6 |
| | | 128 | 27.54 | 0.9518 | 0.2673 | 15.6 | | | 32 | 17.71 | 0.6546 | 0.2081 | 21.0 |
| VQ-VAE-2 + **RAQ** | 256 | 2048 | 33.26† | 0.9881† | 0.1097 | **465.2** | VQGAN + **RAQ** | 64 | 512 | 20.84 | 0.7415 | 0.1024 | **311.4** |
| | | 1024 | **32.77** | 0.9865† | **0.1171** | 239.7 | | | 256 | 20.61† | 0.7332 | **0.1079** | 159.7 |
| | | 512 | **32.24** | **0.9847** | **0.1256** | 133.6 | | | 128 | 20.26† | 0.7207† | **0.1159** | 85.8 |
| | | 256 | **31.33** | 0.9809† | **0.1439** | 67.0 | | | 64 | 19.86† | **0.7052** | **0.1283** | **45.0** |
| | | 128 | 30.39† | 0.9771† | 0.1663 | **38.7** | | | 32 | 19.03 | 0.6787 | 0.1554 | **23.1** |
| | | | **SQ-VAE** / CelebA (128 × 128) | | | | | | | **SimVQ** / ImageNet (128 × 128) | | | |
| SQ-VAE | 2048 | – | 32.04 | 0.9167 | 0.0911 | 449.40 | SimVQ | 4096 | – | 29.98 | 0.9109 | 0.1471 | 1667.61 |
| SQ-VAE | 1024 | – | 31.62 | 0.9141 | 0.0986 | 275.69 | SimVQ | 2048 | – | 29.71 | 0.9067 | 0.1536 | 987.94 |
| SQ-VAE | 512 | – | 30.96 | 0.9023 | 0.1088 | 149.17 | SimVQ | 1024 | – | 29.30 | 0.8986 | 0.1673 | 565.20 |
| SQ-VAE | 256 | – | 30.40 | 0.8927 | 0.1198 | 86.61 | SimVQ | 512 | – | 28.82 | 0.8882 | 0.1835 | 319.99 |
| SQ-VAE | 128 | – | 29.72 | 0.8786 | 0.1295 | 53.87 | SimVQ | 256 | – | 28.25 | 0.8747 | 0.2035 | 174.69 |
| SQ-VAE | 64 | – | 28.72 | 0.8613 | 0.1512 | 28.11 | SimVQ | 128 | – | 27.68 | 0.8601 | 0.2221 | 91.08 |
| SQ-VAE + **RAQ** | 128 | 2048 | 31.85 | 0.9142 | 0.0927 | **521.19** | SimVQ + **RAQ** | 512 | 4096 | **30.03** | **0.9117** | 0.1488 | **2242.70** |
| | | 1024 | 31.40 | 0.9074 | **0.0973** | 269.29 | | | 2048 | **29.74** | 0.9066 | 0.1555 | **1184.28** |
| | | 512 | 30.82 | 0.8990 | **0.1041** | 156.03 | | | 1024 | **29.32** | **0.8990** | **0.1657** | **634.20** |
| | | 256 | 30.09 | 0.8905 | **0.1156** | 90.36 | | | 512 | 28.81 | 0.8885 | 0.1810 | 334.01 |
| | | 128 | 28.95 | 0.8734 | 0.1352 | **53.93** | | | 256 | 28.19 | 0.8741 | 0.2021 | 174.76 |
| | | 64 | 26.97 | 0.8356 | 0.1738 | 23.66 | | | 128 | 27.15 | 0.8503 | 0.2366 | **92.15** |

while staying within confidence bounds at extremes. For SQ-VAE and SimVQ (bottom blocks), RAQ consistently tracks or exceeds fixed-rate baselines across multiple $\widetilde{K}$. In contrast, random selection and the model-based variant degrade notably once more than half of the codebook is removed. These results indicate that RAQ's benefits extend beyond a single backbone and quantization scheme. Beyond reconstruction metrics, we observe systematically higher perplexity at larger $\widetilde{K}$ with RAQ, indicating more balanced code usage and richer latent capacity. Consistent with (Wu & Flierl, 2020; Takida et al., 2022; Vuong et al., 2023), increased codebook perplexity often correlates with better reconstruction. This improvement is especially evident at larger codebook sizes, where RAQ produces non-degenerate codebooks with broadly balanced usage, not just larger codebooks. This aligns with maximizing entropy in discrete representations and indicates that RAQ activates latent capacity underutilized by vanilla VQs.

In summary, RAQ offers substantial advantages in portability and reduced complexity. By consolidating multiple fixed-rate VQ models into a single adaptable framework, it saves training/storage overhead and simplifies deployment. Although minor trade-offs may appear at certain rates, the combination of flexibility and efficiency makes RAQ attractive across diverse VQ frameworks.

## 5.3 Qualitative Evaluation

For our qualitative evaluation, we visualize a single RAQ model based on VQ-VAE-2 (Razavi et al., 2019) reconstructing three Kodak (Kodak, 1993) images at four adapted codebook sizes (Figure 3). As the rate decreases, RAQ exhibits a smooth, graceful degradation: global shapes and hues remain stable while fine textures progressively become smoother. In the parrot (top row), the wrinkles marked with blue boxes remain sharp down to $\widetilde{K} = 256$. In the sailboat (middle row), the green-boxed sail numbers are clearly visible down to $\widetilde{K} = 256$. On the building facade (bottom row),

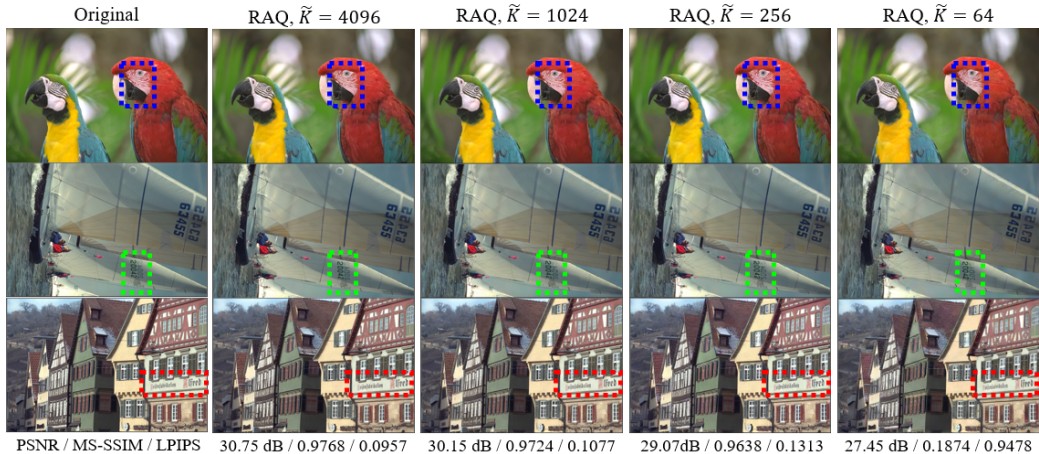

Figure 3: Multi-rate reconstructions with a single RAQ model. Three Kodak ($768 \times 512$) images reconstructed at four codebook sizes. Colored boxes highlight fine details.

the text marked in red boxes maintains readability across all rates. Even at the lowest rate ($\widetilde{K} = 64$), all three images retain coherent structure and plausible colors. This demonstrates the superior effectiveness of our RAQ in handling variable-rate compression tasks, particularly in high-resolution image reconstruction scenarios. Further demonstrations can be found in Appendix A.3.7.

## 5.4 DISCUSSION

**Model-Based RAQ** We also study a post-hoc, model-based variant that adapts rates without re-optimization, though its behavior depends on clustering dynamics. It is generally competitive for codebook reduction ($\widetilde{K} < K$) but shows sensitivity when expanding the codebook ($\widetilde{K} > K$) due to initialization, which can yield unstable assignments and suboptimal local minima (see Appendix A.2.3). In our supplementary study (Appendix A.2.4), plain DKM/$k$-means++/GMM achieve similar performance, whereas *DKM+post-training* closes much of the gap, suggesting a practical fallback when Seq2Seq-based RAQ is infeasible. Despite these limitations, model-based RAQ remains attractive in resource-constrained settings, since it only clusters the existing codebook and scales to large backbones (Yu et al., 2022) where retraining costs dominate.

**Stage-2 Compatibility with Autoregressive Priors** Unlike autoregressive token priors that predict index sequences under a fixed codebook, RAQ creates the codebook itself for any target $\widetilde{K}$ after training, leaving token modeling untouched. For completeness, we pair a single stage-1 RAQ-based VQ model with PixelCNN (van den Oord et al., 2016) and Transformer priors across multiple $\widetilde{K}$ (See Appendix A.3.1). These studies show competitive performance versus fixed-rate baselines while avoiding separate stage-1 retraining per rate, indicating that RAQ maintains stage-2 compatibility and downstream expressivity over multiple $\widetilde{K}$.

**Additional Ablation** Appendix A.3.2 ablates *cross-forcing*. When expanding the codebook ($\widetilde{K} > K$), it stabilizes generation and improves rFID (up to 4.9%) with modest PSNR/SSIM gains (Table 12). For $\widetilde{K} \leq K$, it can slightly underperform, reflecting its expansion-oriented design.

**Computational Cost** Please see Appendix A.2.6 for complexity experiments. RAQ unavoidably increases training time, but because a single model serves multiple rates, the overall cost remains practical. The latency to generate an adapted codebook for a target rate is only 10–140 ms, after which throughput is comparable to fixed-rate VQ models.

**Conclusion** In summary, RAQ enables multi-rate codebook adaptation across VQ models with a single stage-1 backbone while retaining compatibility with stage-2 priors. Remaining limitations include initialization sensitivity in model-based expansion and the need for deeper analysis, which we leave to future work.

**Ethics Statement**   RAQ is designed as a rate-adaptive extension of VQ models and can be applied in all domains where VQ models are used. As with all generative models, attention should be given to potential biases in the training data, as these can affect generated outputs. RAQ does not introduce any new ethical concerns beyond those inherent in VQ models.

**Reproducibility Statement**   Appendix A.1 provides details of the experiments. The complete code necessary to reproduce our experiments is included in the supplementary material.

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

# A  APPENDIX

## A.1  EXPERIMENT DETAILS

### A.1.1  ARCHITECTURES AND HYPERPARAMETERS

The model architecture for this study is based on the conventional VQ-VAE framework outlined in the original VQ-VAE paper (Van Den Oord et al., 2017), and is implemented with reference to the VQ-VAE-2 (Razavi et al., 2019) implementation repositories[1][2][3] and an open-source VQ-VAE + PixelCNN implementation used for our unconditional prior experiments.[4] We also relied on open-source VQGAN resources, including pretrained generator/discriminator weights[5] and a PyTorch implementation[6] used for training and evaluation pipelines.

We are using the ConvResNets from the repositories. These networks consist of convolutional layers, transpose convolutional layers and ResBlocks. Experiments were conducted on two different computer setups: a server with 4 RTX 4090 GPUs and a machine with 2 RTX 3090 GPUs. PyTorch (Paszke et al., 2019), PyTorch Lightning (Falcon, 2019), and the AdamW (Loshchilov & Hutter, 2019) optimizer were used for model implementation and training. Evaluation metrics such as the Structural Similarity Index (SSIM) and the Frechet Inception Distance (rFID) were computed using implementations of pytorch-msssim [7] and pytorch-fid [8], respectively. The detailed model parameters are shown in Table 2. RAQs are constructed based on the described VQ-VAE parameters with additional consideration of each parameter.

Table 2: Architecture and hyperparameters for training VQ-VAE and RAQ models.

| Method | Parameter | CIFAR10 | CelebA | ImageNet |
|---|---|---|---|---|
| VQ-VAE (Van Den Oord et al., 2017) | Input size | $32\times32\times3$ | $64 \times 64 \times 3, 128 \times 128 \times 3$ | $224\times224\times3$ |
| | Latent layers | $8\times8$ | $16\times16, 32\times32$ | $56\times56$ |
| | Hidden units | 128 | 128 | 256 |
| | Residual units | 64 | 64 | 128 |
| | # of ResBlock | 2 | 2 | 2 |
| | Original codebook size ($K$) | $2^4 \sim 2^{10}$ | $2^5 \sim 2^{11}$ | $2^7 \sim 2^{12}$ |
| | Codebook dimension ($d$) | 64 | 64 | 128 |
| | $\beta$ (Commit loss weight) | 0.25 | 0.25 | 0.25 |
| | Weight decay in EMA ($\gamma$) | 0.99 | 0.99 | 0.99 |
| | Batch size | 128 | 128 | 32 |
| | Optimizer | AdamW | AdamW | AdamW |
| | Learning rate | 0.0005 | 0.0005 | 0.0005 |
| | Max. training steps | 195K | 635.5K | 961K |
| **Model-based RAQ** | Original codebook size ($K$) | 64, 128 | 128, 256 | 512 |
| | Adapted codebook size ($\widetilde{K}$) | $2^4 \sim 2^{10}$ | $2^5 \sim 2^{11}$ | $2^6 \sim 2^{12}$ |
| | Max. DKM iteration | 200 | 200 | 200 |
| | Max. IKM iteration | 5000 | 5000 | 5000 |
| | $\tau$ of softmax | 0.01 | 0.01 | 0.01 |
| **RAQ** | Original codebook size ($K$) | 64, 128 | 128, 256 | 512 |
| | Adapted codebook size ($\widetilde{K}$) | $2^4 \sim 2^{10}$ | $2^5 \sim 2^{11}$ | $2^6 \sim 2^{12}$ |
| | Max. Codebook size | 1024 | 2048 | 4096 |
| | Min. Codebook size | 8 | 16 | 64 |
| | Input size (Seq2Seq) | 64 | 64 | 128 |
| | Hidden size (Seq2Seq) | 64 | 64 | 128 |
| | # of recurrent layers (Seq2Seq) | 2 | 2 | 2 |

### A.1.2  DATASETS AND PREPROCESSING

For the **CIFAR10** dataset, the training set is preprocessed using a combination of random cropping and random horizontal flipping. Specifically, a random crop of size $32 \times 32$ with padding of 4 using

---

[1] https://github.com/mattiasxu/VQVAE-2

[2] https://github.com/rosinality/vq-vae-2-pytorch

[3] https://github.com/EugenHotaj/pytorch-generative

[4] https://github.com/KimRass/VQ-VAE-PixelCNN

[5] https://github.com/aa1234241/vqgan

[6] https://github.com/dome272/VQGAN-pytorch

[7] https://github.com/VainF/pytorch-msssim

[8] https://github.com/mseitzer/pytorch-fid

the 'reflect' padding mode is applied, followed by a random horizontal flip. The validation and test sets are processed by converting the images to tensors without further augmentation. For the **CelebA** dataset, the training set is preprocessed with a series of transformations. The images are resized and center cropped to $64 \times 64$ or $128 \times 128$, normalized, and subjected to random horizontal flipping. A similar preprocessing is applied to the validation set, while the test set is processed without augmentation. For the **ImageNet** dataset, the training set is preprocessed with a series of transformations. The images are resized to $256 \times 256$ and center cropped to $224 \times 224$, normalized, and subjected to random horizontal flipping. A similar preprocessing is applied to the validation set, while the test set is processed without augmentation. These datasets are loaded into PyTorch using the provided data modules, and the corresponding data loaders are configured with the specified batch sizes and learning rate for efficient training (described in Table 2). The datasets are used as input for training, validation, and testing of the VQ-VAE model.

## A.2 MODEL-BASED RAQ: ADDITIONAL DETAILS AND ANALYSES

### A.2.1 CODEBOOK CLUSTERING

In this subsection, we formalize codebook clustering for model-based RAQ and fix notation used by DKM. Given a set of the original codebook representations $\mathbf{e} = \{e_i\}_{i=1}^K$, we aim to partition the $K$ codebook vectors into $\widetilde{K}(\leq K)$ codebook vectors $\tilde{\mathbf{e}} = \{\tilde{e}_i\}_{i=1}^{\widetilde{K}}$. Each codebook vector resides in a $D$-dimensional Euclidean space. Using the codebook assignment function $g(\cdot)$, then $g(e_i) = j$ means $i$-th given codebook assigned $j$-th clustered codebook. Our objective for codebook clustering is to minimize the discrepancy $\mathcal{L}$ between the given codebook $\mathbf{e}$ and clustered codebook $\tilde{\mathbf{e}}$:

$$\underset{\tilde{\mathbf{e}}, g}{\arg\min} \, \mathcal{L}(\mathbf{e}; \tilde{\mathbf{e}}) = \underset{\tilde{\mathbf{e}}, g}{\arg\min} \sum_{i=1}^{\widetilde{K}} ||e_i - \tilde{e}_{g(e_i)}|| \tag{7}$$

with necessary conditions

$$g(e_i) = \underset{j \in 1, 2, \ldots, \widetilde{K}}{\arg\min} ||e_i - \tilde{e}_j||, \quad \tilde{e}_j = \frac{\sum_{i: g(e_i) = j} e_i}{N_j} \tag{8}$$

where $N_j$ is the number of samples assigned to the codebook $\tilde{e}_j$.

### A.2.2 CODEBOOK EXPANSION (VIA IKM)

We now describe codebook expansion for model-based RAQ and the inverse-functional DKM (IKM) procedure used to synthesize a larger codebook from a trained one.

While $k$-means clustering is effective for compressing codebook vectors, it has algorithmic limitations when *adding* new codebook vectors. To address this, we introduce inverse-functional DKM (IKM), which increases the number of codebook vectors by approximating the distribution of a trained codebook. We measure distributional discrepancy via maximum mean discrepancy (MMD) between the base codebook and the clustered, synthesized codebook. MMD is a kernel-based two-sample statistic that quantifies distributional similarity (Gretton et al., 2012).

Given a trained base codebook $\mathbf{e}$ of size $K$, IKM generates an expanded codebook $\tilde{\mathbf{e}}$ of size $\widetilde{K} > K$ as follows:

- Initialize a $d$-dimensional adapted codebook vector $\tilde{\mathbf{e}} = \{\tilde{e}_i\}_{i=1}^{\widetilde{K}}$ as $\tilde{\mathbf{e}} \sim \mathcal{N}(0, d^{-\frac{1}{2}} \boldsymbol{I}_{\widetilde{K}})$
- Cluster $\tilde{\mathbf{e}}$ via the DKM process (equation 6): $g_{\text{DKM}}(\tilde{\mathbf{e}}) = \underset{g_{\text{DKM}}(\tilde{\mathbf{e}})}{\arg\min} \, \mathcal{L}_{\text{DKM}}(\tilde{\mathbf{e}}; g_{\text{DKM}}(\tilde{\mathbf{e}}))$.
- Calculate the MMD between the true original codebook $\mathbf{e}$ and the DKM clustered $g_{\text{DKM}}(\tilde{\mathbf{e}})$.
- Optimize $\tilde{\mathbf{e}}$ to minimize the MMD objective $\mathcal{L}_{\text{IKM}}(\mathbf{e}; \tilde{\mathbf{e}}) = \text{MMD}(\mathbf{e}, g_{\text{DKM}}(\tilde{\mathbf{e}})) + \lambda ||\tilde{\mathbf{e}}||^2$.

where $\lambda$ is the regularization parameter controlling the strength of the L2 regularization term. The IKM process can be summarized as $\tilde{\mathbf{e}} = \underset{\tilde{\mathbf{e}}}{\arg\min} \, \mathcal{L}_{\text{IKM}}(\mathbf{e}; \tilde{\mathbf{e}})$. Since DKM does not block gradient flow, we easily can update the codebook $\tilde{\mathbf{e}}$ using stochastic gradient descent (SGD) as $\tilde{\mathbf{e}} = \tilde{\mathbf{e}} - \eta \nabla \mathcal{L}_{\text{IKM}}(\mathbf{e}, \tilde{\mathbf{e}})$.

### A.2.3 INITIALIZATION SENSITIVITY IN EXPANSION

We observe a marked degradation when increasing the codebook size ($\widetilde{K} > K$) under the clustering-only variant. Empirically, this is primarily driven by initialization in IKM: newly synthesized vectors may be placed far from the manifold spanned by the trained base codebook, making soft assignments unstable and prone to poor local minima. This yields higher run-to-run variance and lower reconstruction quality at large $\widetilde{K}$.

While our primary focus is the novelty and efficacy of the Seq2Seq-based RAQ, a systematic study of more robust initialization strategies for model-based RAQ remains promising future work.

### A.2.4 COMPARISON TO CLUSTERING BASELINES

We compare the model-based RAQ variant against standard clustering/compression baselines on a pre-trained VQ-VAE (trained on CelebA $64 \times 64$) using the base codebook ($K = 1024$). For each target adapted codebook size $\widetilde{K} \in \{512, 256, 128\}$, we adapt the codebook with:

- **DKM**: As described in our main paper.
- $k$-**means++**: An effective initialization method for the $k$-means clustering algorithm.
- **Gaussian Mixture Model (GMM)**: A probabilistic clustering method assuming data points come from a finite mixture of Gaussian distributions.
- **DKM + post-training**: Fine-tuning the original VQ model using the compressed codebook obtained from DKM.

Table 3: Comparison of codebook clustering methods for model-based RAQ. Metrics are PSNR↑ / SSIM↑ / Perplexity↑; mean over 4 seeds.

| Method | Adapted codebook size $\widetilde{K}$ | | | |
| --- | --- | --- | --- | --- |
| | 1024 | 512 | 256 | 128 |
| DKM | 26.41 / 0.8351 / 99.9 | 25.24 / 0.8078 / 52.3 | 24.37 / 0.7720 / 32.5 | 23.20 / 0.7415 / 20.0 |
| $k$-means++ | 26.22 / 0.8400 / 60.8 | 25.28 / 0.8071 / 40.2 | 24.51 / 0.7872 / 29.4 | 23.38 / 0.7486 / 21.3 |
| GMM | 26.19 / 0.8350 / 62.4 | 25.22 / 0.8057 / 40.1 | 24.46 / 0.7814 / 30.0 | 23.51 / 0.7487 / 21.3 |
| **DKM + post-training** | **28.39 / 0.8959 / 180.9** | **27.69 / 0.8842 / 95.0** | **26.97 / 0.8675 / 56.7** | **25.66 / 0.8309 / 28.6** |

Table 3 indicate that basic clustering methods achieve similar codebook reduction performance. However, the differentiable nature of DKM uniquely enables efficient fine-tuning, substantially improving the compression performance within the same model architecture. Thus, while initial clustering performance differences are modest, the capability for post-training underscores the notable advantage and novelty of our proposed RAQ method.

### A.2.5 MODEL COMPLEXITY

In this section, we provide a comparison of model complexity in terms of the total number of trainable parameters for our VQ-based models, both with and without our RAQ module. The following tables list parameter counts for the VQ-VAE and RAQ (our proposed method) variations on (i) CIFAR10 (Table 4), (ii) CelebA (Table 5), and (iii) ImageNet (Table 6).

As shown in the tables below, the addition of RAQ introduces a new Seq2Seq component to facilitate codebook adaptation, resulting in a modest increase in the number of parameters:

- In **VQ-VAE**, whose total parameter count is on the order of a few hundred thousand, the RAQ overhead typically adds $\sim$ 200K+ parameters (e.g., for CIFAR10, the total parameter count increases from about 468K to 732K for $K = 128$).
- In **stage-1 VQGAN**, which already has tens of millions of parameters, the additional parameters introduced by RAQ are less than 1% of the total. For instance, the total grows from 72.0M to approximately 72.6M—a practically negligible difference in large-scale settings.

These observations demonstrate that RAQ remains practical for a variety of model scales and does not incur substantial overhead, even when applied to deeper architectures.

Table 4: Number of parameters for training VQ-VAE and RAQ models on the CIFAR10 dataset.

| Method | Encoder | Decoder | # params Quantizer | Seq2Seq | Total |
|--------|---------|---------|--------------------|---------|-------|
| **VQ-VAE** ($K = 1024$) | 196.3K | 262K | 65.5K | - | 525K |
| **VQ-VAE** ($K = 512$) | 196.3K | 262K | 32.8K | - | 492K |
| **VQ-VAE** ($K = 256$) | 196.3K | 262K | 16.4K | - | 476K |
| **VQ-VAE** ($K = 128$) | 196.3K | 262K | 8.2K | - | 468K |
| **VQ-VAE** ($K = 64$) | 196.3K | 262K | 4.1K | - | 463K |
| **VQ-VAE** ($K = 32$) | 196.3K | 262K | 2.0K | - | 461K |
| **VQ-VAE** ($K = 16$) | 196.3K | 262K | 1.0K | - | 460K |
| **RAQ** ($K = 128$) | 196.3K | 262K | 8.2K | 263.7K | 732K |
| **RAQ** ($K = 64$) | 196.3K | 262K | 4.1K | 263.7K | 728K |

Table 5: Number of parameters for training VQ-VAE and RAQ models on the CelebA dataset.

| Method | Encoder | Decoder | # params Quantizer | Seq2Seq | Total |
|--------|---------|---------|--------------------|---------|-------|
| **VQ-VAE** ($K = 2048$) | 196.3K | 262K | 131K | - | 590K |
| **VQ-VAE** ($K = 1024$) | 196.3K | 262K | 65.5K | - | 525K |
| **VQ-VAE** ($K = 512$) | 196.3K | 262K | 32.8K | - | 492K |
| **VQ-VAE** ($K = 256$) | 196.3K | 262K | 16.4K | - | 476K |
| **VQ-VAE** ($K = 128$) | 196.3K | 262K | 8.2K | - | 468K |
| **VQ-VAE** ($K = 64$) | 196.3K | 262K | 4.1K | - | 463K |
| **VQ-VAE** ($K = 32$) | 196.3K | 262K | 2.0K | - | 461K |
| **RAQ** ($K = 256$) | 196.3K | 262K | 16.4K | 263.7K | 740K |
| **RAQ** ($K = 128$) | 196.3K | 262K | 8.2K | 263.7K | 732K |

Table 6: Number of parameters for training stage-1 VQGAN and RAQ models on the ImageNet dataset.

| Method | # params Quantizer | Seq2Seq | Total |
|--------|--------------------|---------|-------|
| **VQGAN** ($K = 512$) | 65.5K | - | 72.0M |
| **VQGAN** ($K = 256$) | 32.8K | - | 72.0M |
| **VQGAN** ($K = 128$) | 16.4K | - | 72.0M |
| **VQGAN** ($K = 64$) | 8.2K | - | 72.0M |
| **VQGAN** ($K = 32$) | 4.1K | - | 71.9M |
| **RAQ** ($K = 128$) | 16.4K | 610K | 72.6M |

### A.2.6 TRAINING/INFERENCE TIME

**Setup** All timings were measured on a single NVIDIA RTX 3090 with batch size 128. We report per-epoch wall-clock time. We compare a single RAQ model ($K{=}256$) against multiple fixed-rate VQ baselines ($K \in \{64, 256, 1024\}$) and the model-based RAQ.

**Training Time**   As expected, adding the Seq2Seq adapter increases per-epoch time versus a single fixed-rate VQ; however, when multiple bitrates are required, one RAQ model replaces several separately trained VQs, reducing total training and maintenance cost.

Table 7: Training time per epoch on the CelebA train set using an NVIDIA RTX 3090 GPU.

| Method | $K$ | Training time per epoch (s) | # params |
|---|---|---|---|
| **VQ-VAE / Model-based RAQ** | 64 | $18.09 \pm 0.256$ | 463K |
| **VQ-VAE / Model-based RAQ** | 256 | $18.43 \pm 0.10$ | 476K |
| **VQ-VAE / Model-based RAQ** | 1024 | $21.64 \pm 0.11$ | 525K |
| **RAQ** | 256 | $514.97 \pm 8.17$ | 740K |

**Inference Time**   We consider two deployment regimes: (i) a *pessimistic* setting that regenerates the adapted codebook for *every* mini-batch (upper bound on cost), and (ii) a *cached* setting that adapts the codebook *once per target rate* $\widetilde{K}$ and reuses it thereafter (realistic case).

Table 8: Inference time per epoch on CelebA (test set) when *regenerating per mini-batch*.

| Method | $\widetilde{K}$ | Inference time per epoch (s) |
|---|---|---|
| **VQ-VAE** ($K = 64$) | – | $1.86 \pm 0.10$ |
| **VQ-VAE** ($K = 256$) | – | $1.91 \pm 0.12$ |
| **VQ-VAE** ($K = 1024$) | – | $1.86 \pm 0.09$ |
| **Model-based RAQ** ($K = 256$) | 64 | $1.98 \pm 0.09$ |
| **RAQ** ($K = 256$) | 64 | $3.05 \pm 0.11$ |
| **Model-based RAQ** ($K = 256$) | 1024 | $70.91 \pm 11.82$ |
| **RAQ** ($K = 256$) | 1024 | $33.21 \pm 0.27$ |

**One-time Codebook Adaptation Latency**   To reflect practical use, we additionally measure the one-time latency to adapt and cache a codebook for a given $\widetilde{K}$; subsequent batches reuse this codebook at no extra cost. The added latency is on the order of 10-140 ms.

Table 9: One-time adaptation latency per target rate (cached at inference).

| $\widetilde{K}$ | Latency (ms) |
|---|---|
| 64 | $9.67 \pm 10.34$ |
| 128 | $18.08 \pm 9.32$ |
| 256 | $36.00 \pm 9.49$ |
| 512 | $71.63 \pm 10.99$ |
| 1024 | $143.94 \pm 12.63$ |

**Summary**   Table 8 should be interpreted as an upper bound (regenerate-per-batch). In realistic deployments (adapting once per $\widetilde{K}$ and caching) the extra latency (Table 9) is negligible, while a single RAQ model still replaces multiple fixed-rate VQs in storage and operational complexity.

## A.3   ADDITIONAL EXPERIMENTS

### A.3.1   STAGE-2 COMPATIBILITY WITH AUTOREGRESSIVE PRIORS

In our end-to-end setups, the stage-2 VQ model is an autoregressive Transformer prior trained on sequences of VQ code indices produced by the stage-1 VQ encoder. Rather than refining pixels directly, it models the distribution of latent token sequences and generates new index sequences that the VQ decoder then maps back to images. This factorization enables us to pair RAQ-adapted latents

with a fixed stage-2 prior across multiple target rates $\widetilde{K}$ without the need for retraining separate stage-1 models.

*RAQ-VAE* denotes a standard VQ-VAE whose stage-1 quantizer is augmented with our Seq2Seq codebook adaptation module. *RAQGAN* analogously augments a stage-1 VQGAN (generator and discriminator) with the same RAQ module, leaving the adversarial training pipeline unchanged.

**RAQ-VAE with PixelCNN**   First, we evaluated the performance of unconditional image generation by combining the trained Stage-1 autoencoders (VQ-VAE and RAQ-VAE) with a PixelCNN decoder that was trained on the Fashion-MNIST dataset. For each codebook size, we generated 10,000 samples via categorical sampling with a temperature of 1.0, and reported both the Inception Score (IS) and the Fréchet Inception Distance (FID), which are the most widely adopted metrics for generative modelling.

Table 10: Evaluation of unconditional image generation. PixelCNN with stage-1 VQ-VAE encoder (*top*) and single stage-1 RAQ-VAE encoder (*bottom*).

| Method | $\widetilde{K}$ | FID $\downarrow$ | IS $\uparrow$ |
|---|---|---|---|
| VQ-VAE ($K$=256) + PixelCNN | – | 48.88 | 4.04 |
| VQ-VAE ($K$=128) + PixelCNN | – | 53.46 | 3.95 |
| VQ-VAE ($K$=64) + PixelCNN | – | 53.69 | 3.95 |
| VQ-VAE ($K$=32) + PixelCNN | – | 54.58 | 4.14 |
| VQ-VAE ($K$=16) + PixelCNN | – | 58.53 | 3.92 |
| | 256 | 47.96 | 4.09 |
| | 128 | 50.81 | 4.08 |
| **RAQ** ($K$=64) + PixelCNN | 64 | 51.24 | 4.06 |
| | 32 | 52.43 | 3.98 |
| | 16 | 58.47 | 3.96 |

As shown in Table 10, RAQ-VAE combined with PixelCNN achieves competitive, and in some cases slightly improved, FID and IS values across all tested codebook sizes compared to baseline VQ-VAE. This demonstrates that the expressivity of the RAQ-adapted latent variables is well preserved for synthesis purposes.

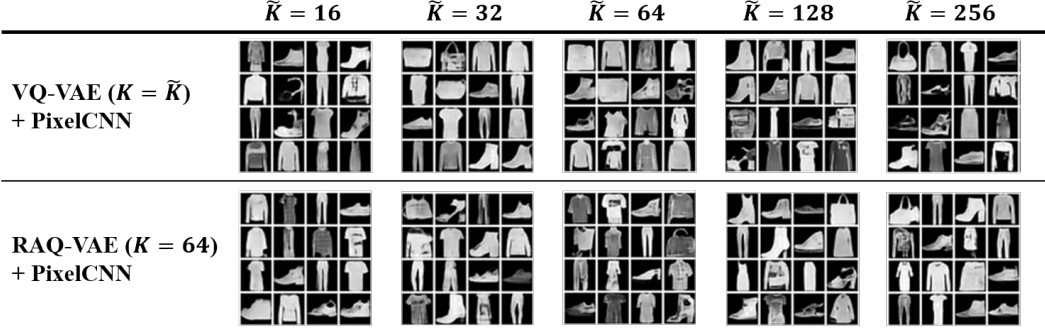

Figure 4: Unconditional samples with a PixelCNN prior. Top: VQ-VAE baselines ($K$=$\widetilde{K}$). Bottom: one RAQ-VAE ($K$=64) with $\widetilde{K}$.

Qualitatively, both fixed-rate VQ-VAE and the single RAQ-VAE produce diverse, coherent samples across the rates. RAQ maintains comparable visual fidelity while varying $\widetilde{K}$ without retraining the stage-1 model.

**RAQGAN with Transformer**   Next, to further evaluate conditional image generation, we integrated VQGAN and RAQGAN stage-1 models with a transformer-based autoregressive prior (minGPT) on the Oxford Flowers102 dataset. We fine-tuned each model (with pretrained encoder, generator, and discriminator) on the Flowers102 train dataset across various codebook sizes and generated 6149 half-conditional test samples.

Table 11: Evaluation of conditional image generation (temperature 1.0, top-$k$ sampling of $k = 100$). MinGPT with stage-1 VQGAN generator (*top*) and single stage-1 RAQGAN generator (*bottom*).

| Method | $\widetilde{K}$ | FID ↓ |
|---|---|---|
| VQGAN ($K$=1024) + minGPT | – | 35.67 |
| VQGAN ($K$=512) + minGPT | – | 40.13 |
| VQGAN ($K$=256) + minGPT | – | 38.47 |
| **RAQ** ($K$=512) + minGPT | 1024 | 34.00 |
| | 512 | 33.47 |
| | 256 | 35.47 |

In Table 11, the results demonstrate that RAQGAN can be effectively combined with an autoregressive transformer prior, yielding performance on par with or superior to standard VQGAN-based models. Furthermore, our results demonstrate that a single RAQ-based model can flexibly support a wide range of adapted codebook sizes with consistently strong generative quality, unlike conventional VQGAN-Transformer approaches that require training separate VQGANs for each bitrate or target rate. Notably, RAQ-adapted latents remain fully compatible with stage-2 autoregressive models, allowing seamless end-to-end synthesis at arbitrary rates. This property is especially beneficial in practical deployment scenarios, where resource constraints or bandwidth conditions may change dynamically, as it enables flexible adaptation to different rates without retraining or model switching.

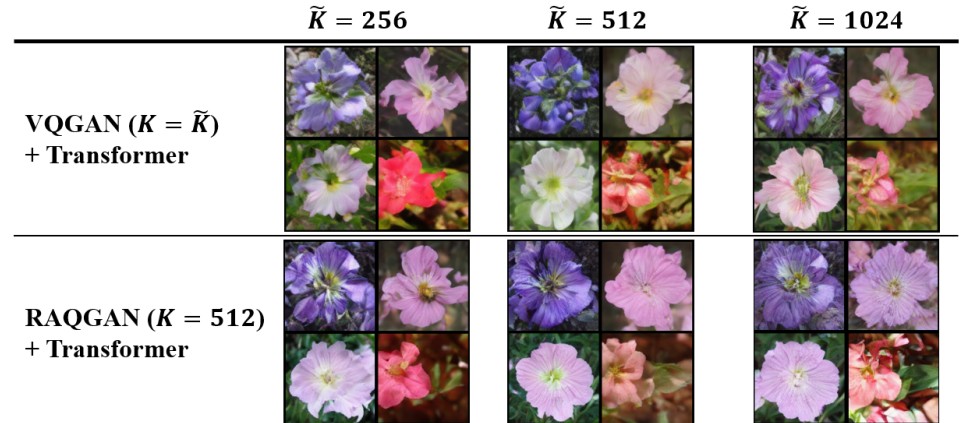

Figure 5: Conditional samples with Transformer (minGPT) on Flowers102 dataset. Top: VQGAN baselines ($K$=$\widetilde{K}$). Bottom: one RAQGAN ($K$=512) with $\widetilde{K}$.

Qualitatively, both VQGAN and the single RAQGAN yield diverse, class-consistent samples. RAQGAN maintains comparable visual quality while varying $\widetilde{K}$ without retraining the stage-1 model.

Across both priors, these small-scale studies suggest that RAQ-adapted latent variables are compatible with stage-2 VQ models over a range of $\widetilde{K}$ and can be paired with off-the-shelf priors without additional stage-1 retraining. While a broader evaluation is left to future work, the evidence here supports straightforward end-to-end use of one stage-1 model under changing bitrate constraints.

### A.3.2 EFFECTIVENESS OF CROSS-FORCING

We performed an ablation study to analyze the impact of our *cross-forcing* training strategy on the stability and fidelity of codebook generation. Using a base RAQ model with an original codebook size $K = 128$ on the CelebA dataset, we compared two variants: (1) *RAQ-w/o-CF* without cross-forcing, and (2) *RAQ-w/-CF* with cross-forcing. Table 12 shows the reconstruction metrics (MSE, PSNR, rFID, and SSIM) for different adapted codebook sizes $\widetilde{K} \in \{32, 64, 128, 256, 512, 1024, 2048\}$. We find that *RAQ-w/-CF* gives significantly better performance than *RAQ-w/o-CF* when $\widetilde{K} > K$, leading to up to $4.9\%$ improvement in rFID and noticeable gains in PSNR and SSIM. In contrast, for smaller or equal codebook sizes ($\widetilde{K} \le K$), *RAQ-w-CF* sometimes

Table 12: Reconstruction performance of RAQ ($K = 128$) **with** or **without** cross-forcing on the CelebA test dataset

| Method | $\widetilde{K}$ | MSE ($\times 10^3$) $\downarrow$ | PSNR $\uparrow$ | rFID $\downarrow$ | SSIM $\uparrow$ |
|---|---|---|---|---|---|
| RAQ-**w/**-CF | 2048 ($\uparrow$) | **1.618$\pm$0.016** | **27.91$\pm$0.04** | **22.64$\pm$0.76** | **0.8810$\pm$0.0013** |
| | 1024 ($\uparrow$) | **1.794$\pm$0.027** | **27.47$\pm$0.07** | **24.67$\pm$0.80** | **0.8710$\pm$0.0016** |
| | 512 ($\uparrow$) | **2.042$\pm$0.021** | **26.90$\pm$0.05** | **26.90$\pm$0.04** | **0.8589$\pm$0.0044** |
| | 256 ($\uparrow$) | **2.412$\pm$0.101** | **26.18$\pm$0.18** | **30.81$\pm$1.59** | 0.8391$\pm$0.0125 |
| | 128 (-) | 2.801$\pm$0.039 | 25.53$\pm$0.06 | 36.30$\pm$1.12 | 0.8209$\pm$0.0072 |
| | 64 ($\downarrow$) | 3.895$\pm$0.095 | 24.10$\pm$0.11 | 47.63$\pm$5.82 | 0.7892$\pm$0.0067 |
| | 32 ($\downarrow$) | **5.357$\pm$0.630** | **22.74$\pm$0.54** | **62.39$\pm$3.76** | **0.7414$\pm$0.0304** |
| RAQ-**w/o**-CF | 2048 ($\uparrow$) | 1.661$\pm$0.056 | 27.80$\pm$0.14 | 23.58$\pm$0.26 | 0.8789$\pm$0.0030 |
| | 1024 ($\uparrow$) | 1.815$\pm$0.050 | 27.42$\pm$0.12 | 25.46$\pm$0.26 | 0.8705$\pm$0.0024 |
| | 512 ($\uparrow$) | 2.068$\pm$0.059 | 26.85$\pm$0.12 | 27.81$\pm$0.42 | 0.8567$\pm$0.0046 |
| | 256 ($\uparrow$) | 2.449$\pm$0.052 | 26.12$\pm$0.09 | 32.32$\pm$1.20 | **0.8407$\pm$0.0031** |
| | 128 (-) | **2.779$\pm$0.015** | **25.57$\pm$0.02** | **36.08$\pm$0.98** | **0.8261$\pm$0.0019** |
| | 64 ($\downarrow$) | **3.860$\pm$0.237** | **24.15$\pm$0.26** | **45.13$\pm$2.79** | **0.7942$\pm$0.0154** |
| | 32 ($\downarrow$) | 6.289$\pm$0.709 | 22.04$\pm$0.47 | 72.85$\pm$16.69 | 0.7338$\pm$0.0225 |

underperforms its counterpart by a small margin. We hypothesize that cross-forcing is specifically designed to stabilize the generation of *larger* adapted codebooks (up to twice the original size), which can result in a slight tradeoff when quantizing at or below the baseline codebook size.

### A.3.3 SEQ2SEQ MODEL SIZE

Regarding the sensitivity of our RAQ to the Seq2Seq model size, we conducted additional experiments. Using our RAQ framework applied to a VQ-VAE-2 (Razavi et al., 2019) baseline on the CelebA (128$\times$128) dataset with an original codebook size of $K = 256$, we compared two configurations:

- **RAQ with 2 LSTM layers:** This configuration uses approximately 528K parameters in the Seq2Seq model (about 10.78% of the total model parameters).
- **RAQ with 4 LSTM layers:** Here, the Seq2Seq model's parameter count increases to approximately 1.06M (about 19.63% of the total model parameters).

Table 13: RAQ performance with different layer configurations on varying codebook sizes. We controlled all other variables over four random seeds (only values significantly outside the confidence interval are bolded).

| Method | $\widetilde{K}$ | PSNR | LPIPS | Perplexity (%) |
|---|---|---|---|---|
| RAQ, 2 layers | 2048 | 33.26 | 0.1097 | 22.71 |
| | 1024 | 32.77 | 0.1171 | 23.41 |
| | 512 | 32.24 | 0.1256 | 26.10 |
| | 256 | 31.33 | 0.1439 | 26.18 |
| | 128 | 30.39 | 0.1663 | 30.21 |
| | 64 | **28.76** | 0.2009 | 37.89 |
| RAQ, 4 layers | 2048 | 33.16 | 0.1052 | 22.53 |
| | 1024 | 32.76 | **0.1107** | 24.28 |
| | 512 | 32.17 | **0.1192** | 26.54 |
| | 256 | **31.49** | **0.1325** | 25.05 |
| | 128 | 30.40 | **0.1548** | 31.36 |
| | 64 | 27.79 | **0.2162** | 35.45 |

The results show that increasing the LSTM layers from 2 to 4 yields only marginal improvements (e.g., a slight improvement in LPIPS) despite nearly doubling the parameter count. These findings indicate that our compact LSTM-based design achieves an appropriate balance between computational efficiency and performance within our current framework. We also note that if the baseline

VQ model were considerably larger (for example, using a ViT-based encoder/decoder), the relative impact of the Seq2Seq model's size might be reduced, and exploring larger architectures could be more promising.

### A.3.4 Codebook Size Switching within a Sequence

To evaluate the stability of our RAQ scheme when switching between codebook sizes within a sequence, we conducted additional experiments on the Kodak dataset. We used a VQ-VAE-2 model trained with RAQ (original codebook size $K = 256$) on ImageNet ($256 \times 256$) and varied the adapted codebook sizes for both the bottom- and top-level latent maps during inference. Table 14 summarizes the PSNR, SSIM, and LPIPS metrics under all combinations of bottom- and top-level codebook sizes.

Table 14: Performance when switching between codebook sizes for top and bottom latent codes on Kodak dataset). The input to the model is 768x512 images of Kodak dataset that is compressed to quantized latent maps of size $192 \times 128$ and $96 \times 64$ for the bottom and top levels, respectively.

| Bottom $\widetilde{K}$ | Top $\widetilde{K}$ | PSNR | SSIM | LPIPS |
|---|---|---|---|---|
| 4096 | 4096 | 30.24 | 0.9739 | 0.10698 |
| 4096 | 1024 | 30.21 | 0.9735 | 0.10850 |
| 1024 | 4096 | 29.80 | 0.9706 | 0.11491 |
| 1024 | 1024 | 29.78 | 0.9701 | 0.11628 |
| 1024 | 256 | 29.70 | 0.9691 | 0.11930 |
| 256 | 1024 | 29.00 | 0.9639 | 0.13077 |
| 256 | 256 | 28.96 | 0.9631 | 0.13298 |
| 256 | 64 | 28.79 | 0.9607 | 0.13941 |
| 64 | 256 | 27.98 | 0.9528 | 0.15973 |
| 64 | 64 | 27.85 | 0.9506 | 0.16575 |

These results confirm that switching codebook sizes within a latent sequence does not degrade reconstruction stability. Prior work reports that the top-level code captures global structure while the bottom-level code encodes local details (Razavi et al., 2019); our findings further reveal that the bottom-level codebook size exerts a stronger influence on reconstruction quality.

### A.3.5 Model-based RAQ

**Rate Reduction**  As analyzed in Section 5.2, RAQ generally outperforms model-based RAQ, but some rate-reduction results on CIFAR10 show that model-based RAQ performs much more stably than in the codebook increasing task. This indicates that simply clustering codebook vectors, without additional neural models like Seq2Seq, can achieve remarkable performance. In Table 15, the performance via codebook clustering was evaluated with different original/adapted codebook sizes $K$: 1024 / $\widetilde{K}$: 512, 256, 128 on CIFAR10 and $K$: 2048 / $\widetilde{K}$: 1024, 512, 256, 128 on CelebA. The conventional VQ-VAE preserved as many codebooks in the original codebook as in the adapted codebook, while randomly codebook-selected VQ-VAE results remained meaningless. Model-based RAQ adopted this baseline VQ-VAE model and performed clustering on the adapted codebook. Model-based RAQ shows a substantial performance difference in terms of reconstructed image distortion and codebook usage compared to randomly codebook-selected VQ-VAE. Even when evaluating absolute performance, it is intuitive that online codebook representation via model-based RAQ provides some performance guarantees.

**Rate Expansion**  In our proposed RAQ scenario, increasing the codebook size beyond the base size is a more demanding and crucial task than reducing it. The crucial step in building RAQ is to achieve higher rates from a fixed model architecture and compression rate, ensuring usability. Therefore, the codebook increasing task was the main challenge. The Seq2Seq decoding algorithm based on cross-forcing is designed with this intention. In Figure 2, the codebook generation performance was evaluated with different original/adapted codebook sizes $K$: 64, 128 / $\widetilde{K}$: 64, 128, 256, 512, 1024 on CIFAR10 and $K$: 128, 256 / $\widetilde{K}$: 128, 256, 512, 1024, 2048 on CelebA datasets. RAQ outperforms model-based RAQ in the rate-increasing task and partially outperforms conventional

Table 15: Reconstruction performances of model-based RAQ for **rate-reduction task** according to adapted codebook size $\widetilde{K}$.

| Method | $\widetilde{K}$ | CIFAR10 ($K = 1024$) | | |
| --- | --- | --- | --- | --- |
| | | PSNR ↑ | rFID ↓ | Perplexity ↑ |
| **VQ-VAE** (baseline model) | - | 25.48 | 51.90 | 708.60 |
| **VQ-VAE** (random select) | 512 | 24.35 | 63.67 | **289.29** |
| | 256 | 22.81 | 78.00 | 111.77 |
| | 128 | 20.87 | 93.57 | 48.87 |
| **Model-based RAQ** | 512 | **24.62** | **55.78** | 285.68 |
| | 256 | **23.81** | **62.53** | **134.54** |
| | 128 | **23.07** | **69.45** | **73.17** |

| Method | $\widetilde{K}$ | CelebA ($K = 2048$) | | |
| --- | --- | --- | --- | --- |
| | | PSNR ↑ | rFID ↓ | Perplexity ↑ |
| **VQ-VAE** (baseline model) | - | 28.26 | 22.89 | 273.47 |
| **VQ-VAE** (random select) | 1024 | 24.02 | 38.92 | **103.50** |
| | 512 | 18.99 | 71.64 | 49.59 |
| | 256 | 23.54 | 115.12 | 27.86 |
| **Model-based RAQ** | 1024 | **26.40** | **31.37** | 102.36 |
| | 512 | **25.24** | **39.07** | **53.45** |
| | 256 | **24.36** | **45.54** | **32.86** |

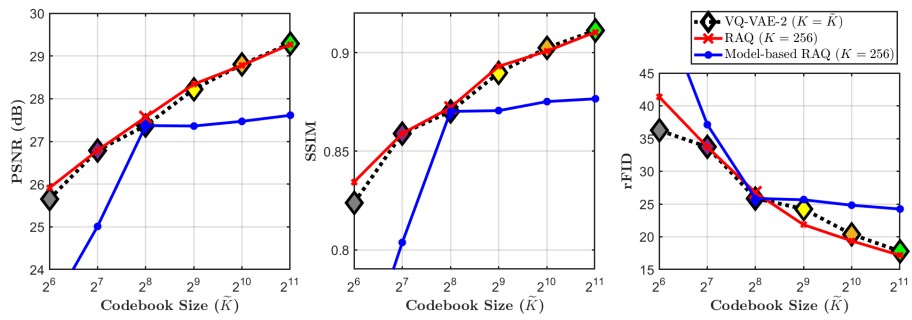

Figure 6: **Reconstruction performance** at different rates (adapted codebook sizes) evaluated on CelebA ($64 \times 64$) test set. In the graph, the black VQ-VAE-2s (Razavi et al., 2019) are separate models trained on each codebook size, while the RAQs are one model per line.

VQ-VAE trained on the same codebook size ($K = \widetilde{K}$). This effect is particularly pronounced on CelebA. However, increasing the difference between the original and adapted codebook sizes leads to a degradation of RAQ performance. This effect is more dramatic for model-based RAQ due to its algorithmic limitations, making its performance less stable at high rates. Improving the performance of model-based RAQ, such as modifying the initialization of the codebook vector, remains a limitation.

### A.3.6 QUANTITATIVE RESULTS

**VQ-VAE** In Table 16 and 17, we present additional quantitative results for the reconstruction on CIFAR10 and CelebA datasets. The error indicates a 95.45% confidence interval based on 4 runs with different training seeds.

**VQ-VAE-2** Figure 6 shows the reconstruction performance using VQ-VAE-2 as the baseline model.

Table 16: **Reconstruction performance** on CIFAR10 dataset. The 95.45% confidence interval is provided based on 4 runs with different training seeds.

| Method | Bit Rate | Codebook Usability | | Distortion | Perceptual Similarity | |
|---|---|---|---|---|---|---|
| | $\widetilde{K}$ | Usage | Perplexity | PSNR | rFID | SSIM |
| **VQ-VAE** $(K=\widetilde{K})$ | 1024 | 972.66±2.97 | 708.60±7.04 | 25.48±0.02 | 51.90±0.51 | 0.8648±0.0005 |
| **VQ-VAE** $(K=\widetilde{K})$ | 512 | 507.52±0.51 | 377.08±5.92 | 24.94±0.01 | 56.65±0.91 | 0.8490±0.0003 |
| **VQ-VAE** $(K=\widetilde{K})$ | 256 | 256±0 | 204.43±4.36 | 24.43±0.02 | 61.40±0.78 | 0.8310±0.0006 |
| **VQ-VAE** $(K=\widetilde{K})$ | 128 | 128±0 | 106.44±1.54 | 23.85±0.01 | 66.70±1.12 | 0.8096±0.0009 |
| **VQ-VAE** $(K=\widetilde{K})$ | 64 | 64±0 | 55.64±0.27 | 23.24±0.01 | 74.00±1.64 | 0.7849±0.0009 |
| **VQ-VAE** $(K=\widetilde{K})$ | 32 | 32±0 | 29.25±0.13 | 22.53±0.02 | 81.68±1.01 | 0.7545±0.0009 |
| **VQ-VAE** $(K=\widetilde{K})$ | 16 | 16±0 | 15.01±0.21 | 21.76±0.01 | 89.75±0.83 | 0.7156±0.0024 |
| **VQ-VAE** $(K=1024)$ (random select) | 1024 | 972.66±2.97 | 708.60±7.04 | 25.48±0.02 | 51.90±0.51 | 0.8648±0.0005 |
| | 512 | 498.38±1.85 | 289.29±16.67 | 24.35±0.11 | 63.67±2.49 | 0.8305±0.0056 |
| | 256 | 253.01±0.66 | 111.77±21.53 | 22.81±0.38 | 78.00±5.07 | 0.7822±0.0100 |
| | 128 | 127.34±0.33 | 48.87±11.31 | 20.87±0.73 | 93.57±9.87 | 0.7254±0.0235 |
| | 64 | 64±0 | 24.31±5.26 | 19.46±0.98 | 109.90±14.20 | 0.6720±0.0309 |
| | 32 | 32±0 | 13.50±1.45 | 17.76±1.12 | 126.57±15.89 | 0.6102±0.0350 |
| **RAQ** $(K=128)$ | 1024 | 979.16±3.72 | 738.48±7.39 | 25.18±0.03 | 54.65±0.99 | 0.8520±0.0007 |
| | 512 | 507.47±0.85 | 387.18±6.87 | 24.82±0.02 | 57.57±0.95 | 0.8417±0.0005 |
| | 256 | 256±0 | 207.78±12.13 | 24.34±0.02 | 61.76±1.22 | 0.8274±0.0008 |
| | 128 | 128±0 | 107.77±0.58 | 23.91±0.0 | 65.37±0.68 | 0.8132±0.0011 |
| | 64 | 64±0 | 55.59±1.81 | 22.87±0.04 | 77.49±2.39 | 0.7770±0.0036 |
| | 32 | 32±0 | 27.77±2.03 | 21.85±0.15 | 89.38±4.33 | 0.7356±0.0064 |
| | 16 | 16±0 | 14.84±0.74 | 20.82±0.09 | 98.93±4.64 | 0.6918±0.0033 |
| **Model-based RAQ** $(K=128)$ | 1024 | 744.36±18.74 | 395.23±2.77 | 24.15±0.03 | 63.88±1.26 | 0.8213±0.0014 |
| | 512 | 430.06±11.58 | 256.23±7.50 | 24.04±0.03 | 64.74±0.96 | 0.8177±0.0012 |
| | 256 | 244.61±3.13 | 185.02±3.31 | 23.93±0.01 | 65.65±1.12 | 0.8139±0.0010 |
| | 128 | 128±0 | 106.44±1.54 | 23.85±0.01 | 66.70±1.12 | 0.8096±0.0009 |
| | 64 | 64±0 | 49.55±1.29 | 22.85±0.55 | 72.61±0.77 | 0.7780±0.0013 |
| | 32 | 32±0 | 25.65±0.76 | 21.88±0.75 | 82.12±1.74 | 0.7405±0.0046 |
| | 16 | 16±0 | 13.79±0.06 | 20.89±0.04 | 95.03±0.34 | 0.6972±0.0010 |
| **RAQ** $(K=64)$ | 1024 | 972.14±6.49 | 725.55±10.90 | 25.04±0.01 | 55.34±1.48 | 0.8487±0.0012 |
| | 512 | 506.38±1.23 | 382.43±10.58 | 24.70±0.02 | 57.91±1.42 | 0.8387±0.0011 |
| | 256 | 255.52±0.48 | 196.17±9.95 | 24.25±0.02 | 61.96±1.00 | 0.8245±0.0012 |
| | 128 | 128±0 | 109.65±3.50 | 23.71±0.01 | 66.89±1.07 | 0.8071±0.0014 |
| | 64 | 64±0 | 56.31±0.46 | 23.23±0.01 | 71.17±1.17 | 0.7897±0.0013 |
| | 32 | 32±0 | 29.62±0.66 | 21.84±0.09 | 90.04±1.44 | 0.7350±0.0038 |
| | 16 | 16±0 | 15.11±0.67 | 20.79±0.18 | 104.86±5.91 | 0.6918±0.0084 |
| **Model-based RAQ** $(K=64)$ | 1024 | 706.20±115.18 | 345.50±107.06 | 23.65±0.13 | 70.30±2.02 | 0.8013±0.0051 |
| | 512 | 428.39±12.29 | 231.41±14.64 | 23.55±0.04 | 71.01±1.38 | 0.7988±0.0005 |
| | 256 | 233.75±4.63 | 140.19±2.82 | 23.39±0.05 | 71.72±1.43 | 0.7935±0.0012 |
| | 128 | 125.07±1.58 | 101.16±16.04 | 23.32±0.05 | 72.68±1.47 | 0.7901±0.0008 |
| | 64 | 64±0 | 55.64±0.27 | 23.24±0.01 | 74.00±1.64 | 0.7849±0.0009 |
| | 32 | 32±0 | 26.21±0.95 | 22.07±0.13 | 81.61±2.26 | 0.7569±0.0014 |
| | 16 | 16±0 | 13.59±0.85 | 20.88±0.23 | 92.84±3.30 | 0.7004±0.0063 |

Table 17: **Reconstruction performance** on CelebA dataset. The 95.45% confidence interval is provided based on 4 runs with different training seeds.

| Method | Bit Rate | Codebook Usability | | Distortion | Perceptual Similarity | |
|---|---|---|---|---|---|---|
| | $\widetilde{K}$ | Usage | Perplexity | PSNR | rFID | SSIM |
| **VQ-VAE** $(K = \widetilde{K})$ | 2048 | 779.07±8.35 | 273.47±6.86 | 28.26±0.03 | 22.89±0.71 | 0.8890±0.0027 |
| **VQ-VAE** $(K = \widetilde{K})$ | 1024 | 456.86±3.53 | 160.35±2.73 | 27.73±0.05 | 26.67±1.43 | 0.8763±0.0029 |
| **VQ-VAE** $(K = \widetilde{K})$ | 512 | 259.59±3.99 | 95.09±1.28 | 27.11±0.01 | 29.77±0.95 | 0.8636±0.0022 |
| **VQ-VAE** $(K = \widetilde{K})$ | 256 | 144.44±2.49 | 57.86±0.91 | 26.46±0.03 | 31.53±1.01 | 0.8481±0.0009 |
| **VQ-VAE** $(K = \widetilde{K})$ | 128 | 80.26±0.99 | 34.98±0.39 | 25.72±0.04 | 36.25±0.98 | 0.8279±0.0027 |
| **VQ-VAE** $(K = \widetilde{K})$ | 64 | 44.94±1.03 | 20.04±0.37 | 24.78±0.03 | 41.22±0.77 | 0.7986±0.0037 |
| **VQ-VAE** $(K = \widetilde{K})$ | 32 | 25.48±0.69 | 12.69±0.31 | 23.76±0.06 | 46.56±1.97 | 0.7660±0.0032 |
| **VQ-VAE** $(K = 2048)$ (random select) | 2048 | 779.07±8.35 | 273.47±6.86 | 28.26±0.03 | 22.89±0.71 | 0.8890±0.0027 |
| | 1024 | 384.31±6.76 | 103.50±3.28 | 24.02±1.10 | 38.92±3.27 | 0.7963±0.0201 |
| | 512 | 210.69±9.23 | 49.59±4.54 | 18.99±1.40 | 71.64±8.27 | 0.7037±0.0221 |
| | 256 | 115.33±7.73 | 27.86±3.39 | 16.33±0.61 | 115.12±11.93 | 0.6353±0.0173 |
| **RAQ** $(K = 256)$ | 2048 | 885.53±6.76 | 347.99±5.17 | 27.96±0.14 | 23.02±0.33 | 0.8858±0.0033 |
| | 1024 | 490.86±4.98 | 187.33±10.37 | 27.51±0.13 | 25.08±0.23 | 0.8758±0.0036 |
| | 512 | 275.84±1.72 | 104.61±5.00 | 26.95±0.086 | 27.96±0.49 | 0.8637±0.0045 |
| | 256 | 144.79±1.21 | 52.63±0.28 | 26.29±0.054 | 32.34±0.86 | 0.8463±0.0030 |
| | 128 | 80.21±4.27 | 32.23±3.87 | 25.13±0.26 | 39.67±2.29 | 0.8162±0.0071 |
| | 64 | 42.93±1.61 | 20.85±1.22 | 24.09±0.21 | 51.57±6.66 | 0.7912±0.0094 |
| | 32 | 22.76±1.57 | 12.32±0.91 | 22.62±0.27 | 69.65±9.49 | 0.7479±0.0129 |
| **Model-based RAQ** $(K = 256)$ | 2048 | 704.17±108.04 | 117.53±33.57 | 26.54±0.10 | 30.34±1.39 | 0.8507±0.0041 |
| | 1024 | 460.77±26.98 | 134.48±11.26 | 26.59±0.06 | 30.49±1.10 | 0.8509±0.0021 |
| | 512 | 279.53±9.48 | 100.64±8.94 | 26.40±0.08 | 30.95±0.98 | 0.8488±0.0017 |
| | 256 | 144.44±2.49 | 57.86±0.91 | 26.46±0.03 | 31.53±1.01 | 0.8481±0.0009 |
| | 128 | 75.31±3.09 | 25.05±1.95 | 24.44±0.25 | 38.95±2.91 | 0.7890±0.0141 |
| | 64 | 41.66±1.22 | 14.73±0.56 | 22.85±0.36 | 48.96±1.13 | 0.7391±0.0192 |
| | 32 | 22.96±0.90 | 10.16±0.95 | 21.81±0.45 | 62.46±0.00 | 0.7077±0.0195 |
| **RAQ** $(K = 128)$ | 2048 | 891.13±7.11 | 345.25±5.15 | 27.91±0.04 | 22.64±0.76 | 0.8810±0.0013 |
| | 1024 | 490.15±14.39 | 176.71±6.19 | 27.47±0.07 | 24.67±0.80 | 0.8710±0.0016 |
| | 512 | 272.60±2.08 | 96.87±2.68 | 26.90±0.05 | 26.90±0.04 | 0.8589±0.0044 |
| | 256 | 152.65±2.45 | 60.90±2.18 | 26.18±0.18 | 30.81±1.59 | 0.8391±0.0125 |
| | 128 | 79.17±0.93 | 31.36±0.77 | 25.53±0.06 | 36.30±1.12 | 0.8209±0.0072 |
| | 64 | 42.71±1.66 | 19.78±2.31 | 24.10±0.11 | 47.63±5.82 | 0.7892±0.0067 |
| | 32 | 22.42±1.92 | 11.43±2.14 | 22.74±0.54 | 62.39±3.76 | 0.7414±0.0304 |
| **Model-based RAQ** $(K = 128)$ | 2048 | 350.02±100.57 | 64.87±21.22 | 22.77±0.78 | 52.37±10.94 | 0.7463±0.0347 |
| | 1024 | 432.15±45.80 | 102.79±17.34 | 25.57±0.19 | 35.62±1.46 | 0.8296±0.0062 |
| | 512 | 262.78±29.47 | 75.63±12.04 | 25.50±0.29 | 36.82±0.73 | 0.8265±0.0026 |
| | 256 | 153.16±5.46 | 53.22±4.62 | 25.42±0.28 | 36.78±1.27 | 0.8285±0.0022 |
| | 128 | 80.26±0.99 | 34.98±0.39 | 25.72±0.04 | 36.25±0.98 | 0.8279±0.0027 |
| | 64 | 41.88±0.72 | 16.70±0.43 | 23.63±0.16 | 47.09±4.09 | 0.7736±0.0080 |
| | 32 | 23.31±0.89 | 9.56±0.77 | 21.64±0.13 | 64.85±6.92 | 0.7037±0.0102 |

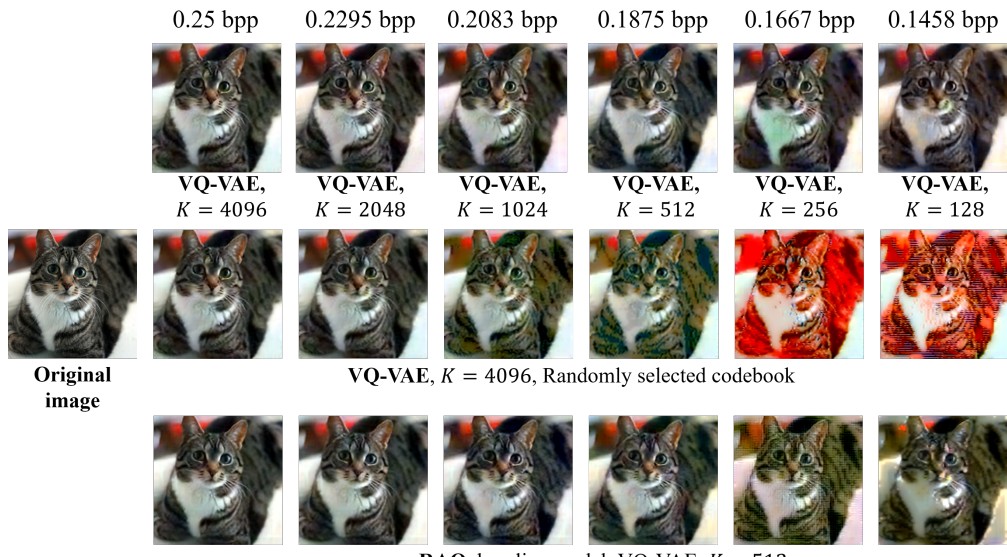

Figure 7: **Qualitative comparison** on ImageNet ($256 \times 256$) at different compression rates. **Top row**: Fixed-rate VQ-VAEs trained separately at each rate. **Middle row**: A single VQ-VAE ($K = 4096$) with randomly selected codebooks. **Bottom row**: Our RAQ with VQ-VAE ($K = 512$) with adapting the codebook size.

### A.3.7 ADDITIONAL QUALITATIVE RESULTS

For our qualitative evaluation, We first compare a single RAQ-based model against multiple VQ-VAEs trained at different rates (0.1458 bpp to 0.25 bpp) on ImageNet ($256 \times 256$). As illustrated in Figure 7, each fixed-rate VQ-VAE (top row) shows a progressive decline in image quality as the rate decreases, consistent with the quantitative evaluation. Unlike RAQ-based reconstruction, randomly selecting codebooks from a single VQ-VAE trained at $K = 4096$ (middle row) results in color distortions and inconsistent hues, especially at 0.1667 bpp. Despite retaining the basic structure, the mismatched usage of codebooks still produces unnatural appearances. By contrast, our RAQ-based VQ-VAE (bottom row), trained at a low-rate base codebook of 0.1875 bpp (roughly $K = 512$), effectively preserves high-level semantic features and color fidelity using only a single model. Notably, it recovers finer details (e.g., the cat's whiskers) far better than models relying on randomly selected codebooks. Although image quality declines slightly at the lowest bpp, largely due to the limited capacity of the baseline VQ-VAE, this issue can be mitigated by using more advanced VQ architectures or refining training procedures. Training RAQ with a smaller original codebook size $K$ can also help reduce performance degradation at lower rates.

We conducted additional experiments using the VQ-VAE-2 model (Razavi et al., 2019) with an original codebook size of $K = 512$. To enhance perceptual quality, we incorporated the LPIPS loss (Zhang et al., 2018) into the training objective and trained the model on the ImageNet dataset at a resolution of $256 \times 256$. The reconstruction task involved reconstructing 24 high-quality images from the Kodak dataset (Kodak, 1993), each with a resolution of $768 \times 512$. For codebook adaptation, we adjusted the codebook size to $\widetilde{K} \in \{4096, 1024, 256, 64\}$ using our RAQ framework. The qualitative results are illustrated in Figure 8. Contrary to Figure 7, where reducing the codebook size in a less complex VQ-VAE model led to noticeable performance degradation, our RAQ-based VQ-VAE-2 demonstrated robust performance across various codebook sizes. Specifically, even as the codebook size decreased, the RAQ-based VQ-VAE-2 model effectively preserved image quality at higher resolutions. These results indicate that increasing the model complexity and refining the training methodology significantly enhance the RAQ framework's ability to adapt codebook rates without compromising reconstruction fidelity.

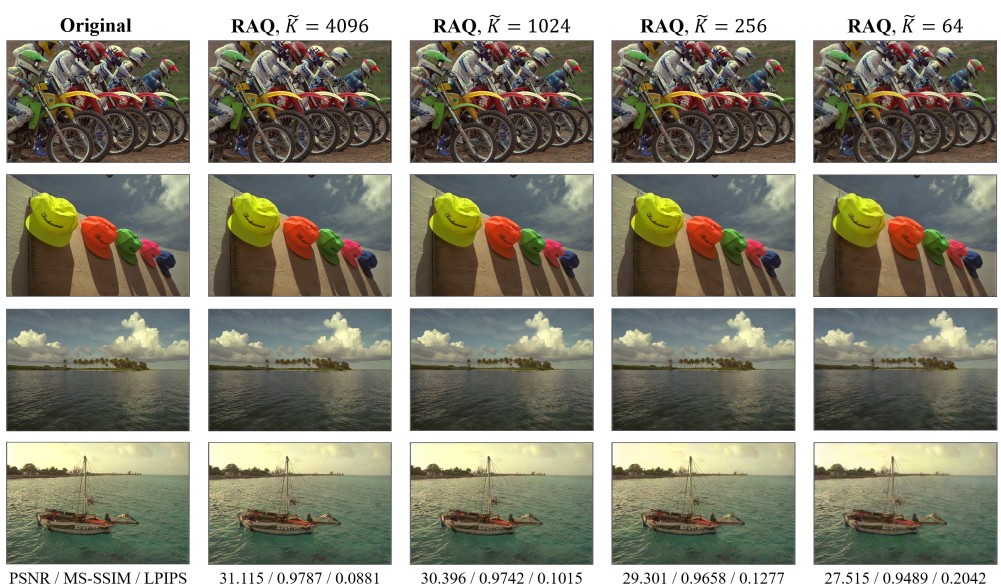

Figure 8: **Reconstructed images** for Kodak (Kodak, 1993) dataset at different rates.

