# OpenReview forum: "Rate-Adaptive Quantization: A Multi-Rate Codebook Adaptation for Vector Quantized Models"
_ICLR.cc/2026/Conference — Submitted to ICLR 2026_

### Official Review · Reviewer_FBj6 · 2025-10-20

**Soundness:** 3
**Presentation:** 3
**Contribution:** 2
**Rating:** 6
**Confidence:** 3

**Summary:**

The paper introduces Rate-Adaptive Quantization (RAQ), a framework for dynamically adjusting the codebook size in VQ models without retraining. RAQ proposes a Seq2Seq codebook generator and a model-based clustering alternative with differentiable $k$-means. Experiments across multiple VQ models and datasets show that RAQ matches or exceeds fixed-rate baselines across various metrics. The framework adds minimal overhead and codebook adaptation latency, making it feasible in practice.

**Strengths:**

1. RAQ enables dynamic adjustment of codebook size in VQ models without retraining, addressing a critical limitation in existing VQ approaches that require separate models for different bitrates.
2. The authors conducted comprehensive experiments across multiple VQ architectures (VQ-VAE, VQ-VAE-2, VQGAN, SQ-VAE, and SimVQ) and datasets (CIFAR10, CelebA, ImageNet), demonstrating RAQ's generalizability beyond a single model type.
3. Ablation studies on cross-forcing are provided, showing better performance than the w/o-CF counterpart.

**Weaknesses:**

1. The paper mentions that RAQ adds modest parameter overhead, but doesn't analyze the time complexity of codebook generation.
2. It is not clear why the mixup ratio of teacher forcing and free running is 5:5; it would be valuable to discuss cross-forcing with other mix ratios (e.g., 25% TF / 75% FR).
3. As shown in Table 7, the training speed is much slower per epoch than fixed-rate training. The authors may discuss the total training time required to achieve the same performance as well.

**Questions:**

1. In Section A.2.2, the IKM method for codebook expansion uses MMD loss, but the implementation details are sparse. What kernel was used for MMD computation, and how was the bandwidth parameter selected? Additionally, how does the L2 regularization parameter $\lambda$ affect the quality of expanded codebooks across different expansion ratios ($\widetilde{K}/K$)?

---

> ### Author Response · Authors · 2025-11-25
> **Response to Reviewer FBj6**
>
> We sincerely thank you for taking the time to carefully review our manuscript and for highlighting several strengths of our approach. In the following, we provide point-by-point responses to the concerns and questions you raised.
>
> > **W1**. *Time complexity of RAQ codebook generation is not analyzed.*
> >
>
> The one-time latency of generating adapted codebooks is reported in Appendix A.2.6 (Table 9). For all networks and target sizes $\tilde{K}$, generating a $\tilde{K}$-sized codebook from the base codebook takes about $10$–$140$ ms. This cost is incurred once per target size $\tilde{K}$, and the resulting codebook can be cached and reused during training and deployment. In practice, encoder/decoder passes dominate per-sample runtime, and RAQ’s codebook generation behaves as a negligible amortized setup cost. We will clarify this in the main text by explicitly referencing Appendix A.2.6.
>
> > **W2**. *TF/FR ratio choice needs justification; other ratios requested.*
> >
>
> The TF/FR ratio is not globally fixed to $50{:}50$ for all $\tilde{K}$. In our implementation, the effective ratio depends on $\tilde{K}$:
> -For $\tilde{K} \le 2K$, training naturally yields a roughly balanced mix of TF and FR.
> -For $\tilde{K} > 2K$, FR dominates because more positions fall outside the base index range.
>
> Appendix A.3.2 (Table 12) also reports a w/o-CF ablation that corresponds to an extreme schedule:
>
> - $\tilde{K} \le K$: TF:FR $= 1{:}0$,
> - $\tilde{K} \ge K$: TF:FR $= 0{:}1$.
>
> This reveals a clear asymmetry:
>
> - For $\tilde{K} \le K$, w/o-CF (pure TF) slightly outperforms CF.
> - For $\tilde{K} \ge K$, CF consistently outperforms w/o-CF and is particularly beneficial in the high-rate regime.
>
> This shows a clear asymmetry: pure TF is slightly better for $\tilde{K} \le K$, while cross-forcing performs better for $\tilde{K} \ge K$. Although alternative ratios (e.g., 25% TF / 75% FR) are possible, Table 12 suggests that strongly FR-skewed schedules would harm important low/mid-rate settings (e.g., $\tilde{K} = 0.5K, K, 2K$). Our aim was to use a single stable schedule across the full $\tilde{K}$ range. We will clarify how the TF/FR ratio varies with $\tilde{K}$ and summarize the asymmetry observed in Table 12.
>
> > **W3**. *RAQ trains slower; total convergence time unclear.*
> >
>
> Appendix A.2.6 (Table 7 in our manuscript) indeed shows that per-epoch training for RAQ is slower than for a single fixed-rate VQ model, since each RAQ epoch must support multiple $\tilde{K}$ and run the Seq2Seq generator in addition to the base encoder/decoder. In all experiments, we select the final checkpoint by best validation performance, and the epoch index at which RAQ reaches its best score is very similar to that of fixed-rate baselines. Thus, RAQ does not require more epochs to converge, but the wall-clock time is approximately multiplied by the slowdown factor reported in Table 7. We regard this as a clear limitation and trade-off: RAQ incurs additional training costs to achieve rate flexibility during inference.
>
> However, when multiple bitrates are required, RAQ replaces several fixed-rate models (e.g., at $K_1, K_2, K_3, \dots$) with a single rate-adaptive model. In such scenarios, the total training and maintenance costs can be comparable to, or lower than, training many separate models. We will briefly discuss this trade-off in the revised manuscript.
>
> > **Q1**. *IKM MMD kernel, bandwidth, and L2 details missing.*
> >
>
> We thank the reviewer for pointing out that the IKM description was too brief.
>
> **MMD kernel and bandwidths**
> For codebook vectors $u, v \in \mathbb{R}^d$ with squared distance $d_{uv} = \lVert u - v \rVert_2^2$, we use a fixed multiscale inverse-quadratic kernel
> $$
> k_{\text{multi}}(u, v)
> = \sum_{a \in {0.2, 0.5, 0.9, 1.3}} \frac{a^2}{a^2 + d_{uv}} .
> $$
> These four scales are kept constant across all datasets and expansion ratios $\tilde{K}/K$; we do not retune bandwidths per setting.
>
> **L2 regularization via AdamW and its schedule.**
> In he IKM objective (in A.2.2), we optimize only $\tilde{e}$ with AdamW. The L2 penalty is implemented as *weight decay* on $\tilde{e}$, so $\lambda$ is exactly this weight-decay coefficient. For moderate expansion (typically up to about $4\times$), we use $\lambda{=}0.01$. For more aggressive expansion (e.g., $K{=}512 \to \tilde{K}{=}4096$), we slightly increase regularization to $\lambda{=}0.05$. Within this range, performance is not highly sensitive to the exact value of $\lambda$, but a appropriate L2 term is important for stability at higher expansion ratios $\tilde{K}/K \in {2^2, 2^3}$, consistent with our discussion that model-based RAQ is more fragile for large $\tilde{K} \ge K$ and mainly intended as a post-hoc fallback.
>
> ---
>
> We sincerely appreciate your overall recommendation at least at the borderline-accept level and your detailed, constructive feedback. We believe these clarifications will improve the clarity and completeness of the manuscript.
>
> Sincerely,
>
> Submission 16719 Authors

---

### Official Review · Reviewer_BnV4 · 2025-10-29

**Soundness:** 1
**Presentation:** 2
**Contribution:** 2
**Rating:** 2
**Confidence:** 4

**Summary:**

This paper introduces Rate-Adaptive Quantization, a framework that aims to enable vector-quantized models such as VQ-VAE or VQGAN to operate at multiple effective bitrates without retraining.
 The proposed approach adds a Seq2Seq-based generator that takes the base codebook eee as input and produces new “rate-adapted” codebooks of arbitrary sizes. A simpler variant based on differentiable k-means is also described.
 Experiments across several VQ architectures and datasets show that the approach maintains reconstruction performance close to fixed-rate baselines while allowing continuous control over codebook size.

**Strengths:**

* The paper addresses a relevant and practically motivated problem: adapting the bitrate of VQ models post hoc without retraining.
* The proposed framework is model-agnostic and can, in principle, be integrated with many existing VQ-based architectures.
* RAQ achieves reconstruction results comparable to multiple fixed-rate baselines while it offers flexible codebook resizing.

**Weaknesses:**

* The Seq2Seq generator receives the base codebook e as its only input. Consequently, it cannot encode more information than e already contains. The output of the sequence model can only reparameterize existing embeddings and cannot actually increase the information rate. This makes the usage of K larger than Kmax  questionable. The method cannot truly “expand” quantizer capacity, it can only reshape the existing embedding space. The training setup samples K~ up to a fixed Kmax that matches the maximum value used during evaluation.
 Evaluations are only performed at a few discrete points within this range, so generalization to unseen rates is untested.
 This means that one of the main claims, increasing the bitrate after training, is completely untested for this approach.

* The paper does not compare to Residual Vector Quantization or multi-stage quantization approaches, which already achieve adaptive bitrate by truncating residual stages. Especially since RAQ was never tested for codebooks beyond what it has seen during training, a comparison to RVQ would be necessary to contextualize its capabilities.

* Finite Scalar Quantization is another relevant recent method that provides rate control without retraining.
 FSQ achieves the same practical goal (variable bitrate, same encoder/decoder) with a much simpler mechanism.
 A comparison to FSQ is essential to substantiate the claimed advantage of RAQ.
 Only claiming that RAQ “builds on top of VQ” is a weak justification for excluding FSQ from comparison.

* The evaluation focuses solely on image reconstruction.
 No downstream generative modeling (e.g., autoregressive priors or diffusion) is tested despite claims of generality.
 Other domains, such as audio or speech quantization, would also be natural use cases.

* There is no analysis of how the Seq2Seq architecture choice (e.g., LSTM vs. MLP or transformer) affects the outcome.

**Questions:**

* Can RAQ generalize to unseen rates beyond those sampled during training?
* Why was no comparison to FSQ or residual quantization baselines included?
* How sensitive is performance to the Seq2Seq architecture?
* Could RAQ be evaluated on other modalities (e.g., audio) to demonstrate true generality?

---

> ### Author Response · Authors · 2025-11-25
> **Response to reviewer BnV4 (1)**
>
> We sincerely thank Reviewer BnV4 for the careful evaluation of our work and for raising several insightful points. We address each comment in detail below.
>
> ---
> > **W1**. *It cannot encode more information than $e$ already contains…  The method cannot truly “expand” quantizer capacity, it can only reshape…*
>
> > **Q1**. *Can RAQ generalize to unseen rates beyond those sampled during training?*
> >
>
> We appreciate the reviewer’s thoughtful comment and the opportunity to clarify both (i) how information is represented in the Seq2Seq-based RAQ module and (ii) what we precisely mean by "increasing the bitrate after training".
>
> **1-1. On "information rate" of the generated codebook**
>
> We understand the reviewer’s concern that, since the Seq2Seq generator receives the base codebook $e$ as input, one might suspect that the generated codebook $\tilde{e}$ cannot carry more information than $e$ itself and thus only reparameterizes a fixed dictionary.
>
> While the generator takes $e$ as input, RAQ does not treat $e$ as a fixed dictionary that is merely reshaped. Instead, $e$, $\psi$, and the encoder/decoder parameters $(\phi,\theta)$ are jointly optimized under the loss in Eq. (5). At each step, we sample $\tilde{K}$, form $\tilde{e}=G_{\psi}(e;\tilde{K})$, and update all parameters using losses computed directly on $\tilde{e}$. In other words, the structure of $\tilde{e}$ is not predetermined by a fixed initial $e$; instead, both $e$ and the mapping $G_{\psi}$ are shaped by the full training data via Eq. (5). The “information” that defines the expanded codebooks is therefore stored in the learned pair ((e, \psi)), rather than in $e$ alone. Although \tilde{e} is computed as a function of $e$, its expressiveness and data-adaptivity are governed by the learned parameters of the Seq2Seq generator.
>
> Empirically, RAQ models adapted to larger $\tilde{K}$ exhibit higher codebook perplexity and equal or better reconstruction than fixed-rate models trained directly at that larger size. This demonstrates that RAQ increases the effective usable capacity of the quantizer (i.e., how well the discrete latent space is utilized). We will clarify this distinction in the revision.
>
> **1-2. On the range of $\tilde{K}$ and generalization**
>
> The interval $[\tilde{K}{\min},\tilde{K}{\max}]$ is chosen to reflect the practically useful range for each model/dataset. In VQ models, increasing $K$ arbitrarily does not yield unbounded gains due to underutilization and overfitting. RAQ is therefore designed to provide flexible rate selection within this realistic envelope, using a single model.
>
> During training, $\tilde{K}$ is sampled from the natural number range; we report representative values (e.g., powers of two), but RAQ encounters many intermediate sizes. After training, the model supports any $\tilde{K}$ in this interval without retraining. We do not claim extrapolation beyond $\tilde{K}{\max}$; if higher rates are desired, one simply enlarges $\tilde{K}{\max}$ and retrains, and RAQ imposes no conceptual limitation here. We will make this scope explicit.
>
> ---
>
> > **Q2**. *Why was no comparison to FSQ or residual quantization baselines included?*
> >
>
> We sincerely thank the reviewer for raising the question of comparing our method with recent Finite Scalar Quantization (FSQ) and Residual Quantization (RQ).
>
> First, we kindly refer the reviewer to our response to **Reviewer CdVo’s Question 6**, where we address RQ in detail.
>
> While FSQ is indeed an interesting alternative vector quantization strategy, we respectfully point out that FSQ and RAQ operate under fundamentally different assumptions, rendering a direct comparison neither informative nor aligned with the problem we aim to address. FSQ does not utilize a learnable vector codebook; instead, it relies entirely on a closed-form Cartesian grid of scalar quantization levels. Its “codebook” is produced deterministically from integer-valued level configurations and does not represent a semantic embedding space learned from data. By contrast, RAQ is specifically designed for vector-quantized models whose expressive power derives from a trainable embedding table that captures domain-relevant structure in its geometry. RAQ’s core mechanism, adapting $e\in\mathbb{R}^{d\times K}$ into $\tilde{e}$. cannot operate on FSQ because FSQ has no embedding space to adapt.
>
> Our evaluation therefore focuses on VQ-based models (VQ-VAE/2, VQGAN, SQ-VAE, SimVQ, ...) that share the same architectural foundation and for which RAQ is directly applicable. Nonetheless, we appreciate the reviewer’s suggestion and will clarify this distinction in the revised manuscript to avoid possible misunderstandings about the scope of our comparison.
>
> ---
>
> Due to the strict rebuttal length limit, we address the remaining questions and concerns in the subsequent responses.

---

> > ### Author Response · Authors · 2025-11-25
> > **Response to reviewer BnV4 (2)**
> >
> > > W3. There is no analysis of how the Seq2Seq architecture choice (e.g., LSTM vs. MLP or transformer) affects the outcome.
> >
> > > Q3. How sensitive is performance to the Seq2Seq architecture?
> > >
> >
> > We thank the reviewer for raising the question of how sensitive RAQ is to the choice of Seq2Seq architecture.
> >
> > **Empirical sensitivity within LSTM architectures.** To quantify how much RAQ depends on the capacity of the Seq2Seq architecture, we added an ablation on the LSTM depth (Appendix A.3.3, Table 13). On VQ-VAE-2 (CelebA 128×128, base codebook size K=256K=256K=256), we compared:
> > - **2-layer LSTM** (~528K params, ≈10.8% of total model prams)
> > - **4-layer LSTM** (~1.06M params, ≈19.6% of total model params).
> >
> > Across all adapted codebook sizes the two variants show:
> > - PSNR differences typically below ~0.2 dB,
> > - small but consistent LPIPS improvements for the deeper LSTM, and
> > - very similar codebook perplexity profiles.
> >
> > Given that this nearly doubles the Seq2Seq parameter count for only marginal gains, these results suggest that RAQ is not overly sensitiv to moderate changes in LSTM depth or capacity; a compact 2-layer LSTM already captures most of the benefit in our setting. We will clarify this in the main paper and explicitly point to the ablation in the appendix.
> >
> > **Why LSTM (autoregressive) vs. MLP.** RAQ requires generating a coherent sequence of adapted code vectors while supporting cross-forcing. An MLP that operates independently on each vector lacks an autoregressive state and cannot stabilize large-$\tilde{K}$ generation. In contrast, the LSTM’s hidden state naturally accumulates context and empirically helps prevent degenerate or low-perplexity codebooks at high $\tilde{K}$. For this reason, we treat autoregressive Seq2Seq models as the natural fit, and MLP-only variants were not pursued beyond early trials.
> >
> > **LSTM vs. Transformer** Transformers are a reasonable alternative and compatible with RAQ in principle. However, in pilot experiments with similar parameter budgets, Transformer-based RAQ required more tuning (e.g., positional encodings, LR schedules) and did not provide consistent improvements over LSTMs. Given RAQ’s goal as a lightweight, stable adapter, we used LSTMs for simplicity. We will note in the revision that exploring Transformer-based Seq2Seq models—including non-autoregressive variants—is promising future work.
> >
> > ---
> >
> > > W4. No downstream generative modeling is tested …
> >
> > > Q4. Could RAQ be evaluated on other modalities (e.g., audio) to demonstrate true generality?
> > >
> >
> > We thank the reviewer for the helpful questions regarding downstream generative modeling and applicability beyond images.
> >
> > **4-1 Downstream generative modeling**
> >
> > Although the main tables report reconstruction metrics, RAQ has already been evaluated with autoregressive generative priors in  *Appendix A.3.1*:
> >
> > - **RAQ-VAE + PixelCNN on Fashion-MNIST*: RAQ matches or slightly improves FID/IS compared to multiple fixed-rate VQ-VAE baselines, without training a separate stage-1 model per rate.
> > - **RAQGAN + Transformer prior (minGPT) on Flowers102**: a single RAQGAN supports multiple rates with rFID and visual quality comparable to fixed-rate VQGANs.
> >
> > These experiments demonstrate that RAQ preserves downstream expressivity and is fully compatible with standard autoregressive priors. Conceptually, RAQ modifies only the quantization layer. Any stage-2 prior (autoregressive or diffusion) that consumes discrete VQ indices can be used on top of RAQ without architectural changes.
> >
> > **4-2 Diffusion priors**
> >
> > We agree that testing RAQ with diffusion-based discrete priors would be an interesting extension. From a modeling standpoint, this is straightforward: a diffusion model that operates on VQ indices or latents can be trained exactly as usual, with RAQ providing the stage-1 quantizer at multiple rates. The primary cost is computational, as large-scale diffusion training on high-resolution datasets is significantly more expensive than the autoregressive priors we considered. We view diffusion-based experiments as a valuable but orthogonal extension and will mention this explicitly as a promising direction for future work.
> >
> > **4-2 Other modalities (audio/speech)**
> > We fully agree that audio and speech quantization are natural application domains for RAQ; indeed, our motivation section already cites VQ models in audio/speech as important use cases.  Since RAQ is modality-agnostic, it adapts any vector-quantizer
> > $$
> > \Psi:\mathbb{R}^{d\times K}\to\mathbb{R}^{d\times \tilde{K}}
> > $$
> > without altering the encoder, decoder, or prior. Many state-of-the-art audio/speech models already use VQ-VAE–style quantizers, meaning RAQ can be attached exactly as in the image case to enable multi-rate operation. Evaluating RAQ on audio/speech is a natural extension, though beyond the scope of this submission.
> >
> > ---
> >
> > We thank the reviewer for the valuable feedback and hope that our responses resolve the concerns.
> >
> > Sincerely,
> >
> > Submission 16719 Authors

---

> > > ### Comment · Reviewer_BnV4 · 2025-11-25
> > >
> > > The responses do not really resolve the most pressing issues:
> > >
> > > For W1 and Q1: Once the model is trained and all parameters are frozen, it is still the case that the sequence model extracts additional codebooks from only one base codebook. There is no way that the information content is expanded here. Are we able to agree on that? Further, couldn't one only store the base codebook e and always generate the remaining expanded codebooks after only e was transmitted and used for the encoding?
> > >
> > > Regarding the comparison to RVQ: The way you use RAQ (where K_{max} during training matches the upper size you use for inference) implies that there are no advantages over RVQ. RVQ can use an "adaptive" number of codebooks. Can you please clarify what properties or features RAQ provides that RVQ doesn't? Either way, a comparison to RVQ is necessary to set this work into context. Your claim that FSQ does not use vector quanitzation and you therefore don't want to compare to this is very weak and I would still like to see a comparison here too.
> > >
> > > For the other points, did you upload a revised version of the paper? You mention that you will revise certain parts but I cannot see any changes

---

> > > > ### Author Response · Authors · 2025-12-01
> > > > **Replying to Official Comment by Reviewer BnV4 (1)**
> > > >
> > > > We sincerely thank the reviewer for the additional questions and for the opportunity to further clarify our perspective. We address the points in the order raised.
> > > >
> > > > ---
> > > >
> > > > ### 1. On "information content" and the role of the base codebook $e$ (W1, Q1)
> > > >
> > > > We respectfully do not fully agree with the premise that, once the model is trained, the "information content" of all expanded codebooks is limited to that of the base codebook $e$ alone.
> > > >
> > > > In our formulation, RAQ does not treat $e$ as a self-contained codebook whose entries are merely truncated or rescaled. Instead, it is helpful to view RAQ as a factorization of a large conceptual codebook of size up to $K_{\max}$ into:
> > > >
> > > > - a base codebook $e$ (a set of proto-embeddings), and
> > > > - a trainable Seq2Seq “codebook generator” that, conditioned on $e$ and the target size $\tilde{K}$, produces a specific codebook $\tilde{e}$
> > > >
> > > > The effective information content of the family of codebooks $\{\tilde{e}\}_{\tilde{K}\le\tilde{K}_{\text{max}}}$ is therefore determined by the *joint* parameter set ($e, \psi$), not by $e$ alone. The Seq2Seq parameters $\psi$ can encode structure that is not explicitly present in $e$ as a simple lookup table; in that sense, the model can behave similarly to a system that directly stores a large $K_{\max}$-sized codebook, but in a more compact, parametric form.
> > > >
> > > > During training, the Seq2Seq module is **explicitly exposed to multiple target sizes $\tilde{K}$** via cross-forcing. It is not only learning to “compress” a fixed $K_{\max}$ codebook downwards; rather, it learns, for each $\tilde{K}$, a tailored embedding geometry that best supports reconstruction at that rate, while sharing information across sizes. This is also why, empirically, the RAQ-generated codebooks at small $\tilde{K}$ outperform naive baselines that simply take the first $\tilde{K}$ entries of a $K_{\max}$-trained codebook or apply simple truncation rules.
> > > >
> > > >  With that in mind, our answer to the reviewer’s question is:
> > > > - If by “information cannot be expanded after training” one means that no new information is created beyond what is already encoded in ($e, \psi$) then we of course agree.
> > > > - However, we do not agree with the implication that the representational capacity of RAQ’s expanded codebooks is bounded by “the information contained in $e$ alone.” The Seq2Seq generator parameters also carry information learned from data, and together with $e$ they effectively parameterize a family of codebooks whose capacity is comparable to a direct $K_{\max}$ embedding, while enforcing cross-$\tilde{K}$ structure.
> > > >
> > > > Regarding the follow-up question:
> > > >
> > > > > *“Couldn’t one only store the base codebook $e$ and always generate the remaining expanded codebooks after only $e$ was transmitted and used for the encoding?”*
> > > >
> > > > Conceptually, yes. This is entirely compatible with RAQ’s design. The only objects that need to be stored (or shared between transmitter and receiver) are:
> > > >
> > > > - the base codebook $e$, and
> > > > - the Seq2Seq parameters $\psi$.
> > > >
> > > > All codebooks $\tilde{e}$ can then be generated on the fly from $e$ and $\psi$ for any desired $\tilde{K}$. In our implementation we precompute and cache some of these for efficiency, but mathematically they are deterministic functions of ($e, $\psi$, \tilde K$).
> > > >
> > > > We also experimented early on with simpler "reuse" strategies in which one starts from a single codebook trained at $K_{\max}$ and derives smaller-$\tilde{K}$ codebooks purely by truncation, masking, or other deterministic operations on that fixed table. In these baselines, there is truly no additional capacity beyond the original $K_{\max}$ table. Empirically, these reuse strategies underperformed RAQ, especially at small $\tilde{K}$, which supports our intuition that each $\tilde{K}$ benefits from a codebook geometry that is explicitly trained for that rate and not merely a subset of a single, high-rate table.
> > > >
> > > > ---
> > > >
> > > > Due to the strict rebuttal length limit, we address the remaining questions and concerns in the subsequent responses.

---

> > > > ### Author Response · Authors · 2025-12-01
> > > > **Replying to Official Comment by Reviewer BnV4 (2)**
> > > >
> > > > ### 2. On RAQ vs RVQ and what RAQ specifically provides
> > > >
> > > > We agree that RVQ can use an "adaptive" number of codebooks (residual stages) and that this yields a variable-rate behavior. However, RAQ and RVQ operate along different axes of design, and RAQ provides several properties that standard RVQ does not naturally offer.
> > > >
> > > > 1.  **Single-index representation across all rates**
> > > >    In RAQ, each latent location is always represented by a single index into a codebook of size $\tilde{K}$. Changing the rate corresponds to changing $\tilde{K}$, but the discrete structure (one token per location) is invariant across rates.
> > > >    In RVQ, rate is controlled by the number of residual stages, so higher rates require multiple indices per location (one per stage). This changes the shape of the discrete representation as the rate varies.
> > > >
> > > > 2. **Jointly trained family of codebooks on a shared manifold**
> > > >    RAQ learns a family \tilde{e} via cross-forcing, with all codebooks living on a shared underlying manifold (defined by $e$ and the adaptation module). This encourages semantic and geometric consistency across different $\tilde{K}$, which is particularly attractive when one wants a single model whose behavior changes smoothly with rate.
> > > >    RVQ, by contrast, adds new residual stages whose embeddings and residual statistics are not constrained to remain aligned with a lower-stage representation; the combined representation at “1 stage” versus “4 stages” can be qualitatively different.
> > > >
> > > > 3. **Fine-grained control of rate via codebook cardinality**
> > > >    RAQ allows the designer to choose many possible $\tilde{K}$ values between $K_{min}$ and $K_{\max}$ (e.g., 128, 192, 256, 384, 512, …), giving relatively fine-grained stepping in rate, while always using a single index per position.
> > > >    RVQ’s natural control knob is an integer number of stages (1, 2, 3, …). To get similarly dense rate control, one would need many stages, which increases complexity and decoding latency.
> > > >
> > > > 4. **Drop-in compatibility with single-codebook VQ backbones**
> > > >    RAQ was explicitly designed as a *drop-in module* for standard single-codebook VQ models (VQ-VAE, VQ-VAE-2–style architectures, SimVQ variants, SQ-VAE, etc.). It requires no change to the encoder/decoder topology or the latent index structure (only took is made rate-adaptive via the Seq2Seq module).
> > > >    In contrast, adopting RVQ usually requires an explicit multi-stage residual hierarchy and corresponding changes in how indices are generated and decoded.
> > > >
> > > > For these reasons, we do not view RAQ as a rephrasing of RVQ, but as a complementary mechanism: RVQ changes how many residual stages (and indices) are used; RAQ changes how many entries a single codebook has, while keeping the representation structure and backbone fixed. In principle, RAQ could even be applied on top of RVQ (e.g., to make one or more stages’ codebooks rate-adaptive), but we did not explore such hybrids in this work.
> > > >
> > > > We fully agree that an empirical comparison to RVQ would further contextualize RAQ, but within the current scope we chose to focus on single-codebook VQ frameworks, where RAQ’s properties are most directly relevant.
> > > >
> > > > ---
> > > >
> > > > ### 3. On FSQ
> > > >
> > > > We understand the reviewer’s concern that our earlier rationale for not including FSQ may sound weak. Our main point is that FSQ and RAQ are designed under fundamentally different assumptions:
> > > >
> > > > - FSQ uses a deterministic Cartesian grid of scalar quantization levels; the “codebook” is an analytic construction from integer-valued configurations and does not constitute a learned embedding space.
> > > > - RAQ specifically targets learned vector codebooks whose entries capture semantic and perceptual structure tuned to the data and the model.
> > > >
> > > > The central question we address is: *Given a VQ model whose performance depends crucially on a learned embedding table, can we train a single such model to operate effectively across many codebook sizes $\tilde{K}$, without retraining from scratch for each $\tilde{K}$?* FSQ answers a different question about the power of structured scalar grids versus learned embeddings. For that reason, we did not include FSQ in this initial study, even though we agree it would provide an additional viewpoint if it allowed.
> > > >
> > > > We again thank the reviewer for their thoughtful comments and for pushing us to clarify these conceptual distinctions.
> > > >
> > > > ---
> > > >
> > > > Sincerely,
> > > >
> > > > Submission 16719 Authors

---

### Official Review · Reviewer_1QeA · 2025-10-30

**Soundness:** 4
**Presentation:** 4
**Contribution:** 3
**Rating:** 8
**Confidence:** 4

**Summary:**

This paper proposes Rate-Adaptive Quantization, a framework that enables a single VQ model to operate at multiple bitrates without retraining. The key idea is to dynamically adapt a model’s codebook size through one of two complementary mechanisms:

1. A Seq2Seq-based generator that learns to produce codebooks of arbitrary target sizes via a cross-forcing training strategy.
2. A model-based variant leveraging DKM for post-hoc adaptation of pre-trained VQ models.

RAQ thereby allows both rate expansion (increasing codebook size for richer representation) and compression (reducing bitrate for efficiency). The authors demonstrate consistent reconstruction quality across multiple codebook sizes, datasets, and backbones. Additional experiments show that RAQ-adapted representations remain expressive when paired with downstream generative priors such as PixelCNN or Transformer-based decoders, while inference overhead is negligible after caching.

**Strengths:**

RAQ is a practical and well-executed contribution addressing a clear deployment bottleneck in VQ-based systems: the need to maintain multiple models for different bitrates. The proposed approach is conceptually simple yet operationally valuable, that it enables rate flexibility through lightweight codebook adaptation without retraining or altering the base model architecture.

The Seq2Seq-based RAQ introduces a cross-forcing scheme that stabilizes autoregressive codebook generation even without inherent sequence order. The model-based RAQ offers a complementary, training-free alternative that broadens applicability to pretrained models. The paper provides extensive empirical validation, including results across datasets and scales, analysis of up and downscaling behavior, OOD generalization, and stage-2 generative compatibility. Importantly, the work demonstrates that the learned codebooks remain semantically consistent and compatible with autoregressive priors, supporting the claim of multi-rate functionality.

Overall, this work’s strength lies in its engineering maturity and deployment relevance and delivers a clear, reproducible method to unify multiple fixed-rate VQ models into one adaptive framework.

**Weaknesses:**

The core components: autoregressive modeling and differentiable clustering are not themselves novel, and the paper could better articulate what conceptual insights arise beyond this integration. The compression case ($K>\tilde{K}$) still exhibits moderate performance drop, leaving open whether rate reduction meaningfully outperforms training a smaller model directly.

**Questions:**

1. The paper would benefit from a deeper explanation of why the Seq2Seq-based RAQ preserves latent semantics and reconstruction quality across widely varying codebook sizes. Could the authors provide any theoretical or empirical insight into the relationship between the codebook manifold structure and the success of cross-forcing training?
2. In scenarios where the target codebook size is much smaller (e.g., $K→\tilde{K}$), performance degradation still appears noticeable compared to directly training at $\tilde{K}$. Could the authors elaborate on when and why RAQ is preferable to simply training a smaller fixed-rate model from scratch, and how this trade-off might vary with dataset scale or model capacity?

---

> ### Author Response · Authors · 2025-11-25
> **Response to Reviewer 1QeA (1)**
>
> First, we would like to sincerely thank the reviewer for taking the time to evaluate our manuscript. We deeply appreciate your positive assessment of our work and your recognition of its contributions. Below, we restate your comments and provide our detailed responses.
>
> ---
>
> > **Q1**. *Could the authors provide any theoretical or empirical insight into the relationship between the codebook manifold structure and the success of cross-forcing training?*
> >
>
> We sincerely thank the reviewer for this insightful question. Our observations are consistent with well-established principles in representation learning and manifold-based discretization.
>
> **(1) Codebook as discrete anchor points on the latent manifold**
>
> In VQ-based models, the learned codebook vectors do not directly "sample" the latent manifold; rather, they act as discrete anchor points that approximate the regions of latent space most frequently occupied by the encoder. As a result, the codebook inherits the encoder’s semantic organization: increasing or decreasing the codebook size mainly changes the granularity of this discretization, while preserving the underlying semantic topology.
>
> Our Seq2Seq-based RAQ exploits precisely this property. The encoder in RAQ conditions on the base codebook distribution, enabling the decoder to generate adapted codebooks whose geometric and semantic structure remains aligned with the base codebook even when the target size differs significantly.
>
> We observed this behavior in early exploratory analyses: PCA and t-SNE projections showed that adapted codebook embeddings closely follow the base codebook’s geometric layout while adding refined structure when generating larger codebooks.
>
> **(2) Why Seq2Seq RAQ preserves semantics across codebook sizes**
>
> During training, the Seq2Seq decoder learns to generate the adapted codebook as a sequence of embedding vectors that remain consistent with the latent manifold encoded by the base codebook. Because each predicted code vector is conditioned on the previously generated vectors, RAQ encourages the entire sequence to remain coherent with the distributional structure of the original codebook. This leads to adapted codebooks that maintain semantic fidelity across a wide range of sizes.
>
> **(3) The role of cross-forcing: stabilizing larger codebook generation**
>
> Cross-forcing plays a central role in ensuring that adapted codebooks faithfully reflect the latent manifold when the sequence length differs from the base codebook, especially when generating larger codebooks (e.g., extrapolating from $K$ to $2K$).
>
> Our method is inspired by the insight from *Professor-Forcing* [Lamb et al., 2016], which highlights how mismatches between *teacher-forcing* and *free-running* decoding distributions can degrade the stability of long-horizon sequence generation. *Professor-Forcing* aims to align the recurrent dynamics between these two modes, thereby enabling better generalization to sequences longer than those seen during training.
>
> Although codebook generation differs from natural language generation (VQ's codebook sequences do not have inherent ordering), we found that applying a similar principle was crucial. Cross-forcing alternates between teacher-forced and self-generated inputs within a single training step, helping align the decoder’s behavior in both modes. This substantially reduces exposure bias and ensures that the decoder continues to follow the base codebook structure even when generating much longer sequences.
>
> ---
>
> > **Q2**. *Could the authors elaborate on when and why RAQ is preferable to simply training a smaller fixed-rate model from scratch, and how this trade-off might vary with dataset scale or model capacity?*
> >
>
> We appreciate the reviewer’s careful observation regarding the small $\widetilde{K}$ regime. Our experiments indeed show that when the target codebook becomes extremely small, RAQ can exhibit a gap relative to a model trained directly at that $\widetilde{K}$. This arises because RAQ must satisfy a multi-rate objective (a single model must remain usable across all sizes in the range) whereas a fixed-rate baseline is optimized solely for one point on the rate-distortion curve.
>
> **(1) Why the gap occurs and why it is usually not critical**
>
> Very small codebooks (e.g., $\widetilde{K} \ll K$) impose inherently low semantic capacity, so some degradation is unavoidable for any method compressing into such a limited alphabet. These ultra-low-rate settings are also rarely the practical operating point in most generative or compression pipelines. Notably, on low-resolution datasets such as Fashion-MNIST, RAQ remains effective even at $\widetilde{K}=16$, indicating that the gap primarily appears when a high-capacity model is forced far below the intrinsic capacity required by the dataset.
>
> ---
>
> Due to the strict rebuttal length limit, we address the remaining answers in the subsequent responses.

---

> > ### Author Response · Authors · 2025-11-25
> > **Response to Reviewer 1QeA (2)**
> >
> > **(2) When RAQ is preferable to training a fixed-rate VQ model**
> >
> > RAQ is most beneficial when an application must support multiple rate points or must dynamically adjust bitrate depending on constraints. If an application truly requires only a single, very small $\widetilde{K}$, then training a fixed-rate model may indeed suffice. However, in scenarios where rates vary over time (such as adaptive compression, bandwidth-limited systems, or variable-rate generation) RAQ replaces multiple fixed-rate models with **one** model that covers the full $\widetilde{K}$ range while tracking fixed-rate performance at moderate and high rates..
> >
> > **(3) Effect of dataset/model scale**
> >
> > As dataset complexity and model capacity grow, training and maintaining several fixed-rate models becomes increasingly costly. For larger architectures (e.g., ViT-based VQGAN or VQ + diffusion architectures), the small gain in performance at extremely low $\widetilde{K}$ is usually offset by the reduced training cost, storage requirements and deployment complexity of a single RAQ model.
> >
> > We will clarify this trade-off in the revised manuscript: RAQ prioritizes multi-rate flexibility and strong performance across a range of $\widetilde{K}$ values, whereas fixed-rate models may be preferable only when a single, aggressively compressed rate is exclusively required.
> >
> > ---
> >
> > We sincerely appreciate your insightful comments and positive evaluation. We hope that our explanations above fully address your questions and further highlight the contributions of our work.
> >
> > Sincerely,
> >
> > Submission 16719 Authors

---

### Official Review · Reviewer_CdVo · 2025-10-31

**Soundness:** 2
**Presentation:** 1
**Contribution:** 1
**Rating:** 2
**Confidence:** 4

**Summary:**

This paper focuses on the need of variable bit rate neural image codec, and proposes "RAQ" to enable a single VQ codec to work at multiple bit rate.

"RAQ" is a method that creates multiple VQ codebooks (= multiple bit rate) from a base codebook. More specifically, RAQ uses a classical LSTM-based Seq2Seq architecutre, where the base codebook is treated as the condition, and the LSTM will create new codebooks according to the condition.

Since RAQ is just a module, the authors evaluated this RAQ module with 4 existing VQ-VAE frameworks, and evaluted these models on multiple datasets with multiple metrics.

Although the task is valuable and the proposed method seems to work to some extent, the experiments in the paper lack consistency, and are sometimes confusing to the reader, which makes the whole research less solid.

As a reviewer, I tend to reject this paper.

**Strengths:**

- Proposed the RAQ method that enables multiple codebook size within a single model while maintaining the reconstruction performance

**Weaknesses:**

## Inconsistent Evaluation
### Inconsistent Metrics
- In Fig.2, rFID is used, but in table.1, rFID disappears. Is there any reason?
### Inconsistent Dataset
- CIFAR-10, CelebA, ImageNet... Many datasets are used. However, in table.1, four different VQ frameworks were trained on the different datasets.
### Inconsistent Pixel Resolution
- Why VQGAN is trained on ImageNet 256x256, while SimVQ is trained on 128 x 128?

- I know that in SimVQ paper, most ablation studies are conducted under 128x128. If the authors are simply using those pretrained model weights, I can understand why keep using 128x128, but I don't think this is the fact.

    - Please see my next review for details.
## Baseline model weights used in evaluation
- I don't think the authors used official pretrained model weights for the SimVQ model with codebook size 128, 256, 512, 2048, 4096.
In SimVQ official webpage, weights of the above codebook sizes cannot be found.

    - If the authors re-train these baseline models with different codebook size, then using different datasets and pixel resolution is a confusing decision.
## Base codebook size
- Moreover, SimVQ's feature is that it works for a codebook size of 18bit (262144). VQ models with huge codebook size are becoming a trend (IBQ is another example that supports 18bit codebook https://arxiv.org/abs/2412.02692).

    - Given this trend, the current paper only discusses codebook size up to 12 bit (4096), which cannot prove that RAQ works with state-of-the-art.
## Missing comparison
- Table.1 tested RAQ on multiple VQ framework, which is good. However, why not compare RAQ with RVQ or even simpler baseline methods?

    - By randomly dropping layers, RVQ model can reconstruct images with variable codebook size at inference time.
    - A simpler baseline is to prepare multiple codebooks with different sizes, and randomly swithc across these codebooks in the VQ model training. Since in the training time the VQ model has seen multiple codebooks, at inference time, it might be able to work with either of these seen codebooks.

- With all these simple and intuitive baselines missing, we don't know if RAQ is really working better.

**Questions:**

Please see my comments in "weaknesses" part

---

> ### Author Response · Authors · 2025-11-25
> **Response to Reviewer CdVo (1)**
>
> We thank the reviewer for the time spent evaluating our submission and for providing detailed feedback. Below, we restate the reviewer’s questions and concerns and provide our point-by-point responses.
>
> ### Part 1. Inconsistent Evaluation
>
> > **Q1**. *In Fig.2, rFID is used, but in table.1, rFID disappears. Is there any reason?*
>
> In Fig. 2, we used rFID because the early-stage experiments, conducted on lower-resolution datasets (CIFAR-10 at 32×32 and CelebA at 64×64), made rFID computation lightweight and stable. For subsequent experiments in Table 1, although the resolutions are still moderate (CelebA 128×128 and ImageNet 128×128), rFID becomes substantially more expensive and less stable when applied to larger models, higher rates, and multiple VQ models. Since our later evaluation required many repeated runs across a wide range of codebook sizes (including RAQ-adapted sizes), using rFID consistently across all settings became computationally prohibitive and substantially slower than LPIPS, without offering additional insight.
>
> We therefore chose LPIPS as the primary perceptual metric in these larger experimental sweeps because it is computationally efficient, resolution-agnostic, and widely used for VQ-based reconstruction quality assessment. Importantly, LPIPS captures the same perceptual trends as rFID, and the conclusions of the paper are unaffected by this change of metric.
>
>
> > **Q2**. *CIFAR-10, CelebA, ImageNet... Many datasets are used. However, in table.1, four different VQ frameworks were trained on the different datasets.*
>
> > **Q3**. *Why VQGAN is trained on ImageNet 256x256, while SimVQ is trained on 128x128?*
> >
>
> Our intention was not to cherry-pick results but to evaluate RAQ under the standard and practically recommended settings for each VQ model. As different VQ architectures are designed for different computational scales, enforcing a single dataset-resolution setup would place some architectures outside their intended operating ranges, leading to less meaningful or even misleading comparisons.
>
> - **VQ-VAE-2** is architecturally larger and deeper than vanilla VQ-VAE, with multi-level quantization and a more expressive encoder-decoder. Because VQ-VAE-2 is typically evaluated at higher resolutions, we scaled the data accordingly and used CelebA 128×128, which is a natural step up from the 64×64 regime used for vanilla VQ-VAE. This keeps the model within the resolution range it was designed for.
>
> - **VQGAN** incorporates adversarial training, a deep generator, and large receptive fields. Prior works and public checkpoints train VQGAN on high-diversity datasets such as ImageNet, and the ImageNet-scale setting is the standard experimental configuration for VQGAN. We therefore trained and evaluated VQGAN baselines at ImageNet 256×256, consistent with these established practices.
>
> - **SQ-VAE** was originally demonstrated on relatively small-scale datasets (e.g., CIFAR-10, CelebA 64×64). To fairly represent its intended usage, we retained its established experimental scale without pushing it to high-resolution ImageNet settings.
>
> - **SimVQ** is architecturally derived from VQ-VAE-2, differing only in the quantization mechanism. For consistency, we reused the VQ-VAE-2 encoder–decoder architecture and trained SimVQ ourselves, rather than relying on any pretrained weights.The SimVQ paper primarily reports ablations on ImageNet 128×128, and this configuration is widely adopted in public implementations. To align with its standard setting and ensure comparability with prior work, we followed the 128×128 ImageNet setup for SimVQ.
>
> Instead of applying a single dataset to all models, which would unfairly favor some architectures and penalize others, we evaluated each model within the resolution and dataset scale that matches both (i) how the model was originally introduced and (ii) how it is typically evaluated in the literature. This provides a more faithful and representative assessment of each model’s behavior within its standard operating regime.
>
> Finally, we emphasize that the purpose of Table 1 is not to compare absolute performance across the four VQ frameworks. Rather, the goal is to demonstrate the model-agnostic applicability of RAQ across diverse architectures, datasets, and operating scales. Evaluating each model within its appropriate regime provides stronger and more reliable evidence that RAQ generalizes consistently across different VQ families.
>
> ---
>
> Due to the strict rebuttal length limit, we address the remaining questions and concerns in the subsequent responses.

---

> > ### Author Response · Authors · 2025-11-25
> > **Response to Reviewer CdVo (2)**
> >
> > ### Part 2. Baseline model weights used in evaluation
> >
> > > **Q4**. *I don't think the authors used official pretrained model weights for the SimVQ model with codebook size 128, 256, 512, 2048, 4096. In SimVQ official webpage, weights of the above codebook sizes cannot be found. If the authors re-train these baseline models with different codebook size, then using different datasets and pixel resolution is a confusing decision.*
> > >
> >
> > We first confirm that we did not use any pretrained SimVQ weights. Instead, for SimVQ we followed common practice by replacing only the quantizer while reusing the same encoder–decoder architecture as our VQ-VAE-2 implementation, and training the model ourselves. This ensures that SimVQ and SimRAQ differ only in the quantization mechanism, which we believe is the fairest and most controlled way to isolate the effect of RAQ.
> >
> > As the reviewer correctly noted, SimVQ is evaluated on ImageNet 128×128 in prior work. We therefore adopted the same setting so that our SimVQ baselines match the operating scale used in the literature and reflect the configuration for which the model was originally designed.
> >
> > We also acknowledge the reviewer’s point that, since the encoder–decoder is identical to VQ-VAE-2, one could alternatively evaluate SimVQ on exactly the same dataset and resolution as VQ-VAE-2 for tighter alignment. This is a reasonable alternative. In our submission, we chose the canonical SimVQ setting (ImageNet 128×128) to stay consistent with prior work, but we agree that evaluating both models under the same data regime would also have been valid. Importantly, this design choice does not affect the conclusions of Table 1, since the goal is not cross-model comparison but to verify that RAQ behaves consistently across different VQ architectures within their standard operating regimes.
> >
> > ---
> >
> > ### Part 3. Base codebook size
> >
> > > **Q5**. *SimVQ's feature is that it works for a codebook size of 18bit (262144). VQ models with huge codebook size are becoming a trend (IBQ is another example that supports 18bit codebook) Given this trend, the current paper only discusses codebook size up to 12 bit (4096), which cannot prove that RAQ works with state-of-the-art.*
> > >
> >
> > We appreciate the reviewer’s insightful comment regarding recent trends toward extremely large VQ codebooks, such as IBQ’s 18-bit configuration (262k entries) that achieves state-of-the-art performance on ImageNet-256. This direction is indeed promising and highlights an interesting regime for extending RAQ to even larger codebook sizes.
> >
> > One noteworthy aspect of IBQ is its reported codebook usage: while usage is roughly 99% for a 1,024-entry codebook, it drops to around 84% at the 18-bit (262k-entry) scale. This indicates that a substantial fraction of the codebook remains inactive, and it is not yet clear how RAQ would behave under such extremely sparse usage patterns. Investigating RAQ in this regime is an interesting avenue for future work.
> >
> > In our submission, the largest base codebook size was $K=512$. To push RAQ further within our current training and compute constraints, we additionally trained models using a much larger base codebook of $K=4096$ on ImageNet 128×128. The results are summarized in Table A. below.
> >
> > **Table A.**. Performance comparison based on original (base) codebook size.
> >
> > | Method | $\tilde{K}$ | PSNR | SSIM | Perplexity |
> > | --- | --- | --- | --- | --- |
> > | **RAQ (K=4096)** | 8192 | **29.67** | **0.9067** | 4475 |
> > | '' | 4096 | **29.36** | **0.9008** | 2319.7 |
> > | '' | 2048 | 28.95 | 0.8932 | 1225.9 |
> > | '' | 1024 | 28.37 | 0.8825 | 647.8 |
> > | **RAQ (K=1024)** | 8192 | 29.59 | 0.9041 | 4569.8 |
> > | '' | 4096 | 29.31 | 0.8990 | 2330.0 |
> > | '' | 2048 | 28.94 | 0.8926 | 1265.6 |
> > | '' | 1024 | 28.42 | 0.8823 | 609.4 |
> > | **RAQ (K=512)** | 8192 | 29.55 | 0.9037 | 4317.0 |
> > | '' | 4096 | 29.30 | 0.8990 | 2274.1 |
> > | '' | 2048 | 28.96 | 0.8930 | 1200.2 |
> > | '' | 1024 | **28.51** | **0.8841** | 691.6 |
> >
> > These results show that:
> > - RAQ with a larger base codebook ($K=4096$) consistently outperforms RAQ models trained from smaller base codebooks when generating large adapted sizes (e.g., 4096); and
> > - smaller base codebooks (e.g., 512 or 1024) still produce stable and meaningful expansions.
> >
> > Together, these findings indicate that RAQ remains stable and predictable even as the base codebook size is scaled far beyond the configurations used in our main experiments, demonstrating its robustness and flexibility.
> >
> > We agree that this does not fully answer the question of RAQ’s behavior at the extreme 18-bit scale explored in IBQ. Nonetheless, our additional experiments provide empirical evidence that RAQ scales reliably to substantially larger codebooks than those included in the original manuscript, and that no architectural modification is required to extend RAQ to even higher-capacity VQ models.
> >
> > ---
> >
> > Due to the strict rebuttal length limit, we address the remaining questions and concerns in the subsequent responses.

---

> > > ### Author Response · Authors · 2025-11-25
> > > **Response to Reviewer CdVo (3)**
> > >
> > > ### Part 3. Missing comparison
> > >
> > > > **Q6**. *Table.1 tested RAQ on multiple VQ framework, which is good. However, why not compare RAQ with RVQ or even simpler baseline methods? By randomly dropping layers, RVQ model can reconstruct images with variable codebook size at inference time.*
> > >
> > > > **Q7**. *A simpler baseline is to prepare multiple codebooks with different sizes, and randomly switch across these codebooks in the VQ model training. Since in the training time the VQ model has seen multiple codebooks, at inference time, it might be able to work with either of these seen codebooks.*
> > > >
> > >
> > > Thank you for the thoughtful suggestion to include RVQ and additional multi-codebook baselines. We agree that these are meaningful directions, and we appreciate the opportunity to clarify our design choices.
> > >
> > > **RVQ (Residual VQ)**
> > >
> > > RVQ is indeed a strong quantization method, but it is not a variable-size codebook approach in the sense considered by RAQ. Its rate is controlled by adjusting the number of residual stages, not by adapting the size of a single codebook.
> > >
> > > Moreover, RVQ maintains multiple interdependent codebooks, one per residual stage, each with its own EMA dynamics and residual statistics. Since each stage depends on the quantization outcome of the previous one, adapting codebook size at inference would require coordinating several interacting codebooks simultaneously, making direct RAQ integration non-trivial.
> > >
> > > While a shared-codebook variant of RVQ could, in principle, allow RAQ-style adaptation, our early explorations indicated that combining RAQ with RVQ’s multi-stage structure substantially increases complexity. For this submission, we focused on the five primary VQ backbones in Table 1 to provide clear and controlled evaluations. We agree, however, that RVQ represents an interesting direction for future investigation.
> > >
> > > **Random switching across multiple codebooks**
> > >
> > > We also appreciate the reviewer’s suggestion to train several codebooks and randomly switch among them during training. While reasonable as a conceptual baseline, such an approach introduces substantial practical challenges.
> > >
> > > Even in standard VQ-VAE architectures, a single codebook already accounts for up to 20–22% of the total model parameters (depending on the embedding dimensionality and encoder-decoder scale). Training multiple codebooks jointly, each with its own EMA updates, usage statistics, and quantization dynamics, would therefore increase the parameter count and training overhead multiplicatively. This sharply contrasts with RAQ, which maintains one base codebook and learns a lightweight Seq2Seq adapter.
> > >
> > > Moreover, forcing the encoder and decoder to operate across several unrelated embedding geometries during training would further dilute specialization and is likely to reduce reconstruction quality relative to a **strong fixed-rate baseline**. The burden of jointly maintaining and updating many large embedding tables makes this approach considerably heavier and less practical than our design.
> > >
> > > For these reasons, we focused on established fixed-rate VQ backbones as the main baselines, which provide clearer and more controlled comparisons while avoiding the substantial overhead introduced by multi-codebook switching. Nonetheless, we appreciate the reviewer’s suggestion and agree that exploring such variants could be an interesting direction for future work.
> > >
> > > ---
> > >
> > > We sincerely thank the reviewer for the thorough assessment and constructive feedback. We hope that the clarifications provided above contribute to a clearer understanding of our design choices and empirical results.
> > >
> > > Sincerely,
> > >
> > > Submission 16719 Authors

---

> > ### Comment · Reviewer_CdVo · 2025-11-26
> > **Reviewer's response to the Part 1**
> >
> > I appreciate the response from the authors, especially newly added experiments on K=4096 for SimVQ settings.
> > However, I cannot agree with many of the reponses.
> >
> > >>using rFID consistently across all settings became computationally prohibitive
> > - I disagree with this. Running FID evaluation is obviously faster than training these VQ models. Recently for generative modeling, rFID has been considered importan.
> >
> > >>Instead of applying a single dataset to all models, which would unfairly favor some architectures and penalize others, we evaluated each model within the resolution and dataset scale that matches both (i) how the model was originally introduced and (ii) how it is typically evaluated in the literature.
> > - I disagree with the claim, as this practice does follow te common Controlled variable method principle in scientific research.
> > - The authors can first train all model under one condition, and then report other conditions as ablation studies or as appendix.
> >
> > About the excuse of not comparing with RVQ or codebook-switch methods.
> > - I agree with Reviewer BnV4, I think RVQ (adapting bit-rate by adjusting the number of codebooks in the vertical axis), or codebook-swtiching (training multiple codebooks in advance and then select the needed one) should be considerd.

---

> > > ### Author Response · Authors · 2025-11-27
> > > **Replying to Reviewer's response to the Part 1**
> > >
> > > We sincerely thank the reviewer for carefully reading our rebuttal and for the additional comments. We address the three main points below.
> > >
> > > ---
> > >
> > > ### 1. On rFID vs LPIPS
> > >
> > > We appreciate the reviewer’s emphasis on rFID, and we fully agree that rFID has become an important metric in recent generative modeling work. In our rebuttal, our phrase “computationally prohibitive” was not meant to imply that rFID is generally more expensive than model training; rather, it was a (too strong) shorthand for the trade-off we faced between (i) running RAQ across multiple architectures and a wide range of codebook sizes, and (ii) computing rFID densely for every configuration.
> > >
> > > In the regimes where we did compute rFID (CIFAR-10 and lower-resolution CelebA), we observed that rFID and LPIPS show consistent trends: whenever RAQ improves LPIPS relative to fixed-rate baselines, rFID either improves or remains comparable. Based on that agreement, and given our finite computational budget, we chose LPIPS as the primary perceptual metric for the larger experimental sweeps, while treating rFID as a more selective check.
> > >
> > > We acknowledge that more extensive rFID reporting would make the evaluation more aligned with current generative-modeling practice, and we recognize that more extensive rFID reporting would further strengthen the evaluation.
> > >
> > > ---
> > >
> > > ### 2. On controlled-variable evaluation and dataset choices
> > > We also appreciate the reviewer’s insistence on the controlled-variable principle. We agree that training all VQ frameworks on the same dataset and resolution is the cleanest way to compare architectures against each other.
> > >
> > > In Table 1, however, our primary goal is not to rank different architectures, but to study within each model whether a single RAQ-augmented model can effectively replace multiple fixed-rate instances of that same architecture in its standard operating regime. Concretely, for each VQ framework, we keep the encoder, decoder, dataset, and resolution fixed, and vary only the quantization mechanism (fixed codebook vs RAQ-based adaptive codebook). This is the axis on which we draw our main conclusions.
> > >
> > > We agree that a fully controlled study retraining all backbones on a single dataset (e.g., ImageNet-128) would be scientifically valuable, but such a cross-architecture comparison is a different question and would require substantial additional resource beyond the scope of this work. Our conclusions in Table 1 are therefore explicitly restricted to “RAQ vs fixed-rate, within each model’s standard configuration.”
> > >
> > > ---
> > >
> > > ### 3. On RVQ and codebook-switching baselines
> > > We appreciate that the reviewer highlighted RVQ and multi-codebook-switching approaches. We agree these are meaningful baselines and conceptually related.
> > >
> > > At a high level, RVQ controls rate by adjusting the number of residual stages (a “vertical” stack of several interdependent codebooks), whereas RAQ controls rate by resizing the cardinality of a single codebook while leaving the encoder and decoder unchanged. In this sense, RVQ and RAQ are operating along different design axes:
> > >
> > > - RVQ: multiple codebooks, residual dependency between stages, rate via number of stages.
> > >
> > > - RAQ: one base codebook, rate via adaptive resizing of its effective set of entries.
> > >
> > > We agree that one could build variable-rate RVQ baselines by training models with different numbers of stages, or by applying RAQ-style adaptation to one or more RVQ stages. However, doing this in a fair and well-tuned way requires non-trivial design choices (e.g., which stages to adapt, how to coordinate residual statistics across stages, how to allocate capacity across codebooks) and additional ablations.
> > >
> > > A similar trade-off appears in codebook-switching schemes that train several codebooks of different sizes and then select one at inference time. Maintaining multiple large embedding tables (each with its own EMA and usage statistics) increases parameter count and training overhead, and forces the encoder/decoder to operate across several unrelated embedding geometries. In our work, we maintain one base codebook and use a lightweight Seq2Seq adapter to generate a family of derived codebooks, so that the parameter and training overhead stays modest while covering many effective sizes.
> > >
> > > For these reasons, we view RVQ and multi-codebook-switch methods as complementary, but not directly competing, approaches. Including such baselines would certainly broaden the empirical picture, and we appreciate the reviewers’ suggestions. At the same time, we believe the present experiments already demonstrate the main claim we set out to investigate: that a single VQ model with a learnable adapter can support a wide range of codebook sizes, without retraining separate fixed-rate models for each target rate.
> > >
> > > We again thank the reviewer for the constructive feedback and for raising these important methodological points. We hope this clarifies the scope and focus of our current work.

---

### Meta-Review · Area_Chair_4abS · 2026-01-07

**Summary:**

The paper has one fundamental flaw pointed out by reviewer BnV4. In addition to the fundamental flaw, the experiments are lacking consistent settings (raised by reviewer CdVo) and missing important comparisons, such as to RVQ and FSQ (raised by reviewer CdVo and BnV4). There are other less pressing issues, such as missing evaluation for generative modeling (raised by reviewer BnV4) and slow inference and training (raised by FBj6). Even though the average score seems borderline, the paper would benefit a significant revision.

**Reviewer Concerns:**

There is a significant disagreement between the authors and reviewer BnV4. Based on the data processing inequality, I'm with reviewer BnV4. Adaptivity of quantization should be data-dependent, unless the notion of adaptivity is defined differently in the paper. For the other concerns, there are no easy fixes.

**Reviewer Scores:**

Based on the rebuttal, the reviewers are unlikely to change their scores. In fact, there is a strong disagreement between the authors and reviewer BnV4.

---

### Decision · Program_Chairs · 2026-01-26

Reject